# Grounding and Calving Cycle of Mertz Ice Tongue

# Revealed by Shallow Mertz Bank

Xianwei Wang[1,2], David M. Holland[2,3], Xiao Cheng[1,5] and Peng Gong[4,5]

1. State Key Laboratory of Remote Sensing Science, and College of Global Change and Earth System Science,

Beijing Normal University. Beijing 100875, China.

2. Center for Global Sea Level Change, New York University Abu Dhabi. Abu Dhabi, United Arab Emirates.

3. Courant Institute of Mathematical Sciences, New York University. New York 10012, United States of America.

4. Ministry of Education Key Laboratory for Earth System Modeling, and Center for Earth System Science,

Tsinghua University, Beijing, China 100084.

5. Joint Centre for Global Change Studies, Beijing, China.

*Correspondence to*: *wangxianwei0304@163.com*

**Abstract**

A recent study, using remote sensing, provided some evidence that a seafloor shoal
influenced the 2010 calving event of the Mertz Ice Tongue (MIT), by partially grounding the
MIT several years earlier. In this paper, we start by proposing a method to calculate Firn Air
Content (FAC) around Mertz from seafloor-touching icebergs. Our calculations indicate the FAC
around Mertz region as $4.87 \pm 1.31$ m.  We then design an indirect method of using freeboard and
sea level data extracted from ICESat/GLAS, FAC, and relatively accurate seafloor topography to
detect grounding sections of the MIT between 2002 and 2008 and analyze the process of
grounding prior to the calving event. By synthesizing remote sensing data, we point out that the
grounding position was localized northeast of the Mertz ice front close to the Mertz Bank. The
grounding outlines of the tongue caused by the Mertz Bank are extracted as well. From 2002 to
2008, the grounding area increased and the grounding became more pronounced. Additionally,
the ice tongue could not effectively climb over the Mertz Bank in following the upstream ice
flow direction and that is why MIT rotated clockwise after late 2002. Furthermore, we
demonstrate that the area-increasing trend of the MIT changed little after calving (~36 $km^2$/a),
thus allowing us to use remote sensing to estimate the elapsed time until the MIT can reground
on the shoal. This period is approximately 70 years. The calving of MIT can be cyclical because
of the shallow Mertz Bank location and the flow rate of the tongue.  The calving cycle of the
MIT explains the cycle of sea-surface condition change around the Mertz.
**Keywords:** Mertz Ice Tongue, firn air content, iceberg grounding, Mertz Bank, calving cycle.

## 1. Introduction

Surface-warming induced calving or disintegration of floating ice has occurred in Antarctica, such as the Larsen B ice shelf (Scambos et al., 2000, 2003; Domack et al., 2005; Shepherd et al., 2003). While surface or sub-surface melting has largely been recognized to contribute to floating ice loss in Antarctica (Depoorter et al., 2013), calving caused by interaction with the seafloor has not been widely considered. The Mertz Ice Tongue (MIT) was reported to have calved in 2010, subsequent to being rammed by a large iceberg, B-9B (Legresy et al. 2010). After the calving, the areal coverage of the Mertz polynya, and sea-ice production and dense, shelf-water formation in the region changed (Kusahara et al. 2011; Tamura et al. 2012). However, the iceberg collision may have only been an apparent cause of the calving as other factors had not been fully considered such as seafloor interactions (Massom et al., 2015; Wang. 2014). By comparing inverted ice thickness to surrounding bathymetry, and combining remote sensing, Massom et al., (2015) considered that the seabed contact may have held the glacier tongue in place to delay calving by ~8 years. The interaction of the MIT with the seafloor, the exact grounding location of the MIT before calving and how severe the grounding was are still not well-known.

The MIT (66°S-68°S, 144°E-150°E, Fig. 1) is located in King George V Land, East Antarctica, with an ice tongue extending over 140 km from its grounding line to the tongue front and approximately 30 km wide at the front (Legresy et al., 2004). Much field exploration has been conducted around Mertz and the increasing availability over the last decade of remote sensing, hydrographic surveying, and bathymetric data allow the causes of ice tongue instability to gradually come into focus. From satellite altimetry, a modest elevation change rate of 0.03 m/a (Pritchard et al., 2012) and a freeboard change rate of -0.06 m/a (Wang et al., 2014) were found,

which implied that the combined effects of surface accumulation and basal melt were not dramatic for this ice tongue. For the MIT, investigations of tidal effects, surface velocity, rift propagation, and ice front propagation (Berthier et al., 2003; Frezzotti et al., 1998; Legresy et al., 2004; Lescarmontier et al., 2012; Massom et al., 2010, 2015) have been conducted with an objective of detecting underlying factors affecting stability. Grounding as a potential factor can affect the stability of an ice tongue, as recently pointed out by Massom et al. (2015). However, without highly accurate bathymetric data, it is impossible to carry out such study. Fortunately, In 2010, a new and high resolution bathymetry model, for the seafloor surrounding the Mertz, with a resolution of 100 m was released for the Terra Adelie and George V continental margin (Beaman et al., 2011), and incidentally later used to generate the Bedmap-2 (Fretwell et al., 2013). Such accurate data provides an opportunity for better exploring seafloor shoals and their impact on the instability of MIT. In this study, we focus on the grounding event of the MIT from 2002 to 2008. A method for grounding event detection is proposed and the grounding of the MIT before calving is investigated. A calving cycle of the MIT caused by grounding is discussed as well.

**2. Data**

The primary data used to investigate ice tongue grounding in this study are Geoscience Laser Altimeter System (GLAS) data onboard the Ice, Cloud and land Elevation Satellite (ICESat) and the seafloor bathymetry data mentioned above. In this section, ICESat/GLAS and bathymetry data, as well as some preprocessing are introduced.

**2.1 ICESat/GLAS**

The ICESat is the first spacebone laser altimetry satellite orbiting the Earth, lunched by National Aeronautics and Space Administration (NASA) in 2003 (Zwally et al. 2002) with

GLAS as the primary payload onboard. ICESat/GLAS was operated in an orbit of ~600 km and
had a geographical coverage from 86°S to 86°N. ICESat/GLAS usually observed in nadir
viewing geometry and employed laser pulses of both 532 nm and 1064 nm to measure the
distance from the sensor to the ground (Zwally et al. 2002). On the ground, ICESat/GLAS's
footprint covered an area of approximately 70 m in diameter, with each adjacent footprints
spaced by ~170 m. The horizontal location accuracy of the footprint was about 6 m (Abshire et al.
2005). The accuracy and precision of ICESat/GLAS altimetry data were 14 cm and 2 cm
respectively (Shuman et al. 2006). ICESat/GLAS usually made two or three campaigns a year
from 2003 to the end of 2009, with each campaign lasting for about one month. With billions of
laser footprints received by the telescope, 15 types of data were produced for various scientific
applications, named as GLA01, GLA02, … GLA15. In this study, GLA12 data (elevation data
for polar ice sheet) covering the Mertz from release 33 during the interval of 2003 to 2009 is
used, the spatial distribution of which is shown in Fig. 2.
**2.2 Seafloor Topography**

Detailed bathymetry maps are fundamental spatial data for marine science studies

(Beaman et al., 2003, 2011) and crucially needed in the data-sparse Antarctic coastal region
(Massom et al. 2015). Regionally, around Mertz, a large archive of ship track single-beam and
multi-beam bathymetry data from 2000 to 2008 were used to generate a high resolution Digital
Elevation Model (DEM), the spatial coverage of which can be found in Fig. 2 of Beaman et al.
(2011) and bathymetry data coverage over the Mertz region can be found from S-Fig. 1. The
DEM product was reported as having a vertical accuracy of about 11.5 m (500 m depth) and
horizontal accuracy of about 70 m (500 m depth) in the poorest situation (Beaman et al. 2011).
Around Antarctica, seafloor topography data from Bedmap-2 was produced by Fretwell et al.
(2013) which adopted the DEM from Beaman et al. (2011). In this study, Bedmap-2 seafloor
topography data covering Mertz is employed to detect the contact between seafloor and the MIT.
Because of inconsistent elevation systems for ICESat/GLAS and seafloor topography data, the
Earth Gravitational Model 2008 (EGM08) geoid with respect to World Geodetic System 1984
(WGS-84) ellipsoid is taken as reference. Since seafloor topography from Bedmap-2 is
referenced to the so-called g104c geoid, an elevation transformation is required and can be
implemented through Eq. (1).
$$E_{sf} = E_{seafloor} + gl04c_{to\_wgs84} - EGM2008 \qquad\qquad (1)$$
where $E_{sf}$ and $E_{seafloor}$ is the seafloor topography under EGM08 and g104c respectively,
$gl04c_{to\_wgs84}$ is the value needed to convert height relative to gl04c geoid to that under WGS-84,
and *EGM2008* is the geoid undulation with respect to WGS-84.
**3. Methods**
**3.1 Grounding Detection Method**
ICESat/GLAS data has been widely used to determine ice freeboard, or ice thickness,
since its launch in 2003 (Kwok et al., 2007; Wang et al., 2011, 2014; Yi et al., 2011; Zwally et
al., 2002, 2008). To study ice freeboard, draft, and grounding of the MIT through time,
ICESat/GLAS GLA12 data from release 33 from 2003 to 2009 are used as mentioned, and the
spatial coverage of which can be seen in Fig. 2. The methods we designed for grounding
detection of the MIT are now introduced. First, assuming a floating ice tongue, based on
freeboard data extracted in different observation dates, the ice draft of the MIT is inverted. Next,
ice bottom elevation is calculated based on the inverted ice draft and the lowest sea-surface
height. Finally, the ice bottom is compared with seafloor bathymetry and ice grounding is
detected. The underlying logic for grounding detection is that if the inverted ice bottom is lower
than seafloor, we can draw a conclusion that the ice tongue is grounded rather than floating.

The method to extract a freeboard map using ICESat/GLAS from multiple campaigns

over the MIT was described in Wang et al. (2014). Here, we do not revisit it in detail but
introduce it schematically.  Four steps are included in freeboard map production for each of the
datasets from November 14, 2002, March 8, 2004, December 27, 2006 and January 31, 2008..

The first step is on data preprocessing, saturation correction, data quality control, and

tidal correction removal. The magnitude of the ICESat/GLAS waveform can become saturated
because of different gain setting, or the occurrence of clouds. Thus the saturated waveforms with
*i_satElevCorr* (i.e. an attribute from GLA12 data record) greater than or equal to 0.50 m are
ignored and those with *i_satElevCorr* less than 0.50 m are corrected by adding the correction
back (Wang et al. 2012, 2013). Additionally, measurements with *i_reflctUC* greater than or equal
to one are ignored. Furthermore, tidal correction from the TPX07.1 tide model in GLA12 data
record is removed to obtain elevation data on the instantaneous sea surface condition. Finally,
elevation data related to the WGS-84 ellipsoid and EGM 08 geoid for ICESat/GLAS from 2003
to 2009 is prepared for subsequent use.

The second step is to derive sea-level height according to each track and to calculate

freeboard for each campaign. Because of tidal variations near the MIT, surface elevations of the
MIT can vary as well. To derive sea-level height from ICESat/GLAS and provide a reference for
freeboard calculation for different campaigns, ICESat/GLAS data over the MIT within a buffer
region (with 10 km as buffer radius of MIT boundary in 2007) are selected and sea-level height
is determined as the lowest elevation measurement along each track (Wang et al. 2014).
Freeboard is then calculated by subtracting the corresponding sea-level height from elevation
measurement of the MIT according to different tracks in the same campaign. Thus freeboard data
for different campaigns from 2003 to 2009 is obtained.

The third step is to relocate footprints using estimated ice velocity. ICESat observed the

MIT almost repeatedly along different tracks in different campaigns (Fig. 2). However,
observation from only one campaign cannot provide good coverage of the MIT, which drives us
to combine all observations from 2003 to 2009 together to produce a freeboard map of MIT. Fig.
2 shows the spatial coverage of ICESat/GLAS from 2003 to 2009 over the Mertz, but the
geometric relation between tracks is not correct over the MIT because the tongue was fast
moving and observed in different years by the ICESat. The region observed in an earlier
campaign would move downstream later (Wang et al. 2014). For example, ICESat collected data
from track T31 on March 22, 2003 and T165 (Fig. 2) on November 1, 2003 respectively. Fig. 2
shows the distance between track T165 and T31, ~7.5 km without considering ice flow. However
because of the fast moving ice tongue, the distance of their actual ground tracks on the surface of
the MIT should be a little larger because T165 is located upstream and observed later. Thus
footprints relocation using ice velocity is critical to obtain accurate geometric relations among
different tracks. The ice velocity data from Rignot et al. (2011) generated from InSAR data from
2006 to 2010 is used to relocate the footprints of ICESat/GLAS. Thus the correct geospatial
relations between observations from different campaigns can be achieved on November 14, 2002,
March 8, 2004, December 27, 2006, and January 31, 2008, through Eqs. (2) and (3). The
freeboard change with time should be considered as well, but this contribution is neglected
because freeboard comparison from crossing tracks showed a slightly decreasing trend of -0.06
m/a on average (Wang et al. 2014). The spatial distribution of freeboard data over the MIT
corresponding to November 14, 2002, is shown in Fig. 5(a).
$$X = x + \sum_{i=1}^{n} v_{xi}\Delta t + v_{xm}t_m \qquad\qquad (2)$$
$$Y = y + \sum_{i=1}^{n} v_{yi}\Delta t + v_{ym}t_m \qquad (t_m = t_2 - t_1 - n\Delta t) \qquad (3)$$
where $x$ and $y$ are locations in the X and Y directions from ICESat measurement directly;
$X$ and $Y$ are locations in the X and Y directions after relocation; $v_x$ and $v_y$ are the ice velocities in
the X and Y directions respectively; $t_1$ and $t_2$ are the start and end times; $\Delta t$ is the time interval
and $n$ indicates the largest integer time steps for time interval between $t_1$ and $t_2$; $t_m$ is the
residual time;  In this work, $\Delta t$ is set as 10 days; $v_{xi}$ and $v_{yi}$ is derived from ice velocity field
according to different locations during relocation and may change in different time intervals.

The forth step is to interpolate the freeboard map using the relocated freeboard data from

step three. Inverse Distance Weighting, Natural Neighbor, Spline and Kriging are most widely
used interpolation techniques (Childs. 2004). Kriging interpolation under spatial analysis toolbox
of ArcGIS is selected in this study to produce freeboard maps of the MIT because it can provide
an optimal interpolation estimate for a given coordinate location by considering the spatial
relationships of a data set. . With this method, freeboard maps of the MIT are produced on
November 14, 2002, March 8, 2004, December 27, 2006, and January 31, 2008, because of
known ice tongue outlines from Landsat images.

Ice draft is calculated with Eq. (4) assuming hydrostatic equilibrium and the lowest sea-

surface height (further discussed later in Section 6.2.2) is extracted as well from ICESat/GLAS
data from all campaigns covering this region, which was -3.35 m under EGM 08 (WGS-84). For
time varying sea-surface heights caused by tides, the minimum sea-surface height can allow ice
with a given draft to ground to the seafloor. Then, ice bottom elevation is calculated by
considering the ice draft and the lowest sea-surface height. To compare the ice bottom with the
seafloor, an elevation difference of both is calculated. In this way, a negative value indicates that
ice bottom is lower than seafloor, which corresponds to grounding.
$\rho_w D = \rho_i (H_f + D - FAC)$ (4)
where $D$ is ice draft, i.e. vertical distance from sea surface to bottom of ice; $H_f$ is freeboard, i.e.
vertical distance from sea surface to top of snow; $\rho_w$ and $\rho_i$ are densities of ocean water and ice,
respectively. In this study, ice and sea water density are taken as 915 kg/m$^3$ and 1024 kg/m$^3$,
respectively (Wang et al., 2014); $FAC$ is the firn air content, the decrease in thickness (in meters)
that occurs when the firn column is compressed to the density of glacier ice, as defined in
Holland et al., (2011) and Ligtenberg et al. (2014). The calculation of firn air content around
Mertz is introduced in Section 3.2. In this work, we define the elevation of at the underside
(bottom) of the tongue as $E_{ice\_bottom}$ and is calculated by Eq. (5).
$E_{ice\_bottom} = E_{sea\_level} - D$ (5)
where $E_{ice\_bottom}$ corresponds to elevation of the ice bottom. $E_{sea\_level}$ is the lowest sea-surface
height among extracted sea-surface height from different tracks and different campaigns, which
is -3.35 m. Similarly, the elevation difference of ice tongue bottom and seafloor is defined as
$E_{dif}$, which can be calculated by Eq. (6).
$E_{dif} = E_{ice\_bottom} - E_{sf}$ (6)
where $E_{dif}$ is elevation difference by subtracting the seafloor elevation  from the ice bottom.
**3.2. Firn Air Content Estimation Method**

The Antarctic ice sheet is covered by a dry, thick firn layer which represents an

intermediate stage between fresh snow and glacial ice, having varying density from Antarctic
inland to the coast (Van den Broke, 2008). The density and depth of the Antarctic firn layer has
been modeled (e.g., Van den Broke, 2008) using a combination of regional climate model output
and a steady-state firn compaction model. However, for ice thickness inversion, Firn Air Content
(FAC) is usually used to make the calculation convenient (Rignot and Jacobs. 2002) and is
defined as the decrease in thickness (in meters) that occurs when the firn column is compressed
to the density of glacier ice (Holland et al., 2011). Time-dependent FAC has also been modeled
by considering the physical process of the firn layer (e.g., Ligtenberg et al. 2014). For the MIT,
there are some in-situ measurements of snow thickness available from Massom et al. (2010) who
used a snow layer depth of 1 m to derive the thickness of surrounding multi-year, fast sea ice.
However on the surface of the MIT, no in-situ measurements of density or depth of firn layer are
available.

Because of different density and thickness of the firn layer on top of an ice tongue, it is

challenging to simulate the density profile of the MIT without in-situ measurements as control
points.   In this study, we use FAC extracted from adjacent seafloor-touching icebergs to
investigate the grounding of the MIT rather than FAC from modeling. MIT may be composed of
pure ice, water, air, firn or snow that makes ice mass calculation complicated. However, if
assuming a pure ice density only to calculate ice mass, the thickness of MIT must be corrected
by FAC. FAC correction to ice thickness can be inferred from surrounding icebergs calving from
MIT using Eq. (4) when knowing ice draft and freeboard assuming hydrostatic equilibrium.
Thus it is critical to target and use icebergs fulfilling these requirements to solve Eq. (4), such as
slightly grounded icebergs above already known seafloor with observed freeboard.  From Smith
(2011), icebergs can be divided into three categories based on bathymetry and seasonal pack ice
distributions: grounded, constrained, and free-drifting icebergs. Without occurrence of pack ice,
an iceberg can be free-drifting or grounded. Free-drifting icebergs can move several tens of
kilometers per day, such as iceberg A-52 (Smith et al. 2007). Grounded icebergs can be firmly or
lightly anchored. Heavily grounded icebergs have firm contact with the seafloor and can be
stationary for a long time, such as iceberg B-9B (Massom. 2003). However, slightly grounded
icebergs may have little contact with the seafloor and can possibly move slowly under the
influence of ocean tide, ocean currents, or winds, but much slower than free-drifting icebergs.
The relation of grounded and ice drifting velocity is not well-known. However, from slowly
drifting or nearly stationary icebergs in open water, we can determine if an iceberg is grounded.
Because of the heavily grounded iceberg B-9B to the east of the MIT blocking the
drifting of pack ice or icebergs from the east, icebergs located between B-9B and the MIT are
most likely generated from the Mertz or Ninnis glaciers. We calculate the FAC from these
icebergs and later apply it to grounding event detection of the MIT. Around the MIT, the
locations of three icebergs ('A', 'B' and 'C') were identified using MODIS and Landsat images
in austral summer, 2006 and 2008 and shown in Fig. 4. Fortunately, ICESat/GLAS observed
these icebergs on February 23, 2006 (54th day of 2006) and February 18, 2008 (49th day of
2008). This allows us to analyze the behavior of the icebergs three-dimensionally. From Fig. 4a,
icebergs 'A', 'B' and 'C' changed position little in about two months (from 28 to 85 day of
2006).  Thus we can consider these icebergs slightly grounded. These slightly grounded icebergs
may plough the seafloor and leave ridges or grooves. In Pine Island Trough, ridges on the
seafloor have been already found with a range of 1 to 2 m, which was believed to be influenced
by grounding icebergs drifting with tides (Jakobsson et al. 2011; Woodworth-Lynas et al. 1991).
From this viewpoint, we are confident that under the lowest sea level (lowest tide), these iceberg
must be grounded, which means that the ice draft inverted from freeboard measurement
assuming hydrostatic equilibrium must be greater than or equal to water depth.  Based on this
analysis, we can take water depth as draft to calculate the FAC.
Because only 'A' and 'C' were observed by track T1289 of the ICESat/GLAS in 2006,
freeboard and water depth from bathymetry for both are used to calculate the FAC (Fig. 4, 9 and
Table 1). However, the icebergs were not stationary, which indicates only some parts were
grounded. In this study, only the top two largest freeboard measurements of icebergs 'A' and 'C'
from T1289 in 2006 are employed to calculate the FAC with Eq. (7) with a least-squares method
under hydrostatic equilibrium.
$$FAC = H_{f\_k} + D_k - \frac{\rho_w}{\rho_i} D_k + \varepsilon_k \qquad\qquad (7)$$
where $k$ is used to identify different icebergs 'A' or 'C', $H_f$ is the top two largest freeboard
measurement of each iceberg, $D$ is ice draft which is the same as sea water depth and is taken
from seafloor bathymetry directly, $\varepsilon$ is a residual for FAC.
Table 1 shows the freeboard and seafloor bathymetry under the icebergs in 2006 for FAC
calculation and grounding detection of icebergs in 2008 (detailed freeboard values for these
icebergs can be seen from Fig. 9). With freeboard and seafloor measurements from iceberg 'A'
and 'C' in 2006 (Table 1), the FAC is calculated as about $4.87 \pm 1.31$ m. Two icebergs 'A' and 'B'
were observed by the same track T1289 of the ICESat/GLAS on February 18, 2008 and thus are
used to evaluate the grounding detection using this FAC. From iceberg trajectories observed by
remote sensing (Fig. 4b), we know, iceberg 'A' drifted away from its original position. Thus it
was not grounded.  However, iceberg 'B' kept rotating in this period without drifting away, from
which we can consider it grounded. Such grounding status determined from remote sensing can
also be detected with our method since the elevation difference of ice bottom and seafloor from
Table 1 does clearly indicate a grounded iceberg 'B' and a floating iceberg 'A'. Thus, our FAC
estimation works well around Mertz.
**4. Accuracy of Grounding Detection**

The accuracy of $E_{dif}$ is critical to grounding detection of the MIT. From Eq. (1) to (6),

we   find different components of the error sources, such as from sea surface height
determination, ice draft, seafloor bathymetry, and elevation transformation. Meanwhile,
uncertainty of ice draft is primarily determined by that of freeboard and $FAC$. Furthermore, the
uncertainty of freeboard is influenced by footprint relocation and freeboard changing rates.
Considering all mentioned above, the error source of elevation difference $E_{dif}$ can be
synthesized by Eq. (8):
$\Delta E_{dif} = \Delta E_{sl} + a(\Delta H_f + \Delta E_{re} + \Delta E_{fb_c} + \Delta FAC + \Delta E_{krig}) + \Delta E_{sf} + \Delta E_{trans}$         (8)
where $a = \frac{\rho_i}{\rho_w - \rho_i}$; $\Delta$ stands for error of each variable; $\Delta E_{dif}$ stands for error of final elevation
difference of ice bottom and seafloor; $\Delta E_{sl}$, $\Delta H_f$, $\Delta E_{re}$, $\Delta E_{fb\_c}$, $\Delta FAC$, $\Delta E_{sf}$, $\Delta E_{krig}$, and
$\Delta E_{trans}$ stand for errors caused by sea surface height extraction, freeboard extraction, freeboard
relocation, freeboard changing rates, FAC calculation, seafloor bathymetry, kriging interpolation
and elevation system transformation, respectively.

Usually, the influence of elevation system transformation on final elevation difference

can be neglected. Based on the error propagation law, the uncertainty of elevation difference $E_{dif}$
can be described by Eq. (9):
$\varepsilon E_{dif} = \sqrt{(\varepsilon E_{sl})^2 + a^2[(\varepsilon H_f)^2 + (\varepsilon E_{re})^2 + (\varepsilon E_{fb_c})^2 + (\varepsilon FAC)^2 + (\varepsilon E_{krig})^2] + (\varepsilon E_{sf})^2}$(9)
where $\varepsilon$ indicates the uncertainty of each parameter.

**4.1 Uncertainty of kriging interpolation**

Fig. 5a shows the spatial distribution of freeboard data over the MIT used for detecting

grounding on November 14, 2002. The spatial difference of ICESat/GLAS between Fig. 2 and
Fig. 5 are caused by footprint relocation, after which the spatial geometry between different
tracks is reasonably correct. In the lower right of the Mertz ice front (Fig.  5a), the freeboard
distance between track T1289 and T165 is about 7 km.  In these data gaps, freeboard data used
for grounding detection in Section 3.1 is interpolated using kriging. Thus, knowing the
uncertainty of kriging interpolation is critical to final grounding detection.

To investigate interpolation uncertainty of the kriging method, freeboard measurements

should be compared with interpolation ones. Thus, a testing region with freeboard measurements
is selected, indicated by a blue dashed square in Fig. 5a, about 7 km$\times$7 km. A freeboard map is
first interpolated with gray dots only (Fig. 5a) using kriging. Then, the freeboard measurements
(284 of green dots in Fig. 5a) are compared with interpolation in the square. The spatial
distribution and the histogram of freeboard difference derived by subtracting krigged freeboard
from freeboard derived from ICESat/GLAS is shown in Fig. 5b.

In this square, the freeboard measurement varies from 31.6 m to 40.0 m with 36.6 m in

average. However, the interpolated freeboard varies from 32.9 m to 39.6 m with 35.9 m in
average. From the freeboard difference results (Fig. 5b), we find that the interpolation results
show similar results compared with freeboard derived from ICESat/GLAS. The interpolated
freeboard has an accuracy of -0.7$\pm$1.8 m. The interpolated freeboard using kriging can reflect
the actual freeboard well. Also, the distribution of freeboard difference in Fig. 5b does not show
obvious geospatial variation trend.
**4.2 Grounding Detection Robustness**

Since sea level is extracted from ICESat/GLAS data track by track, we use $\pm0.15$ m as

the uncertainty of elevation data ($\varepsilon E_{sl}$). Also from Wang et al. (2014), we can see the uncertainty
of freeboard extraction ($\varepsilon H_f$) is $\pm0.50$ m. From Rignot et al. (2011), the error of ice velocity
ranged from 5 m/a to 17 m/a. Assuming that ice velocity varied by 17 m/a (an upper threshold),
the relocation error horizontally could reach ±54 m in an average of three years. Wang et al.
(2014) extracted the average slope of the MIT along ice flow direction as 0.00024. However,
because of large crevasses on the surface, we use 50 times of this value as a conservative
estimate of the average slope. In this way, we can estimate $\varepsilon E_{re}$ as ±0.65 m when considering a
three-year period. The annual rate of freeboard change from 2003 to 2009 is -0.06 m/a (Wang et
al. 2014). Therefore, we consider the freeboard stable over this period. However, when
combining data from different time periods then $\varepsilon E_{fb\_c}$ is estimated as about ±0.18 m if
considering three years time difference. From Beaman et al. (2011), considering elevation
uncertainty at the worst situation when water depth is 500 m, $\varepsilon E_{g104c}$ is ±11.5 m. For kriging
interpolation, from analysis in Section 4.1, 1.8 m is taken as the uncertainty. Using all these
errors above, we calculate the final uncertainty of elevation difference as $\pm 23$ m.
From the calculations above, we can say that $E_{dif}$ less than 23 m corresponds to a very
robust grounding event. However, if the $E_{dif}$ is greater than 23 m, we can not confirm grounding.
$E_{dif}$ in the interval of -23m to 23 m corresponds to slight grounding or floating. We can also
determine different contributions of each separate factor to the overall accuracy. Seafloor
bathymetry contributes the largest part and is the dominant factor affecting the accuracy of
grounding detection.
**5. Grounding Detection Results**
The spatial distribution of elevation difference $E_{dif}$ and outlines of the MIT from 2002 to
2008 are shown in Fig. 6. A buffer region with radius of 2 km (region between black and grey
lines in Fig. 6) is introduced to investigate grounding potential of the MIT, if it approached there.
The elevation difference less than 46 m (twice of elevation difference uncertainty $\varepsilon E_{dif}$) both
inside and outside of the outline is extracted and the corresponding statistics are shown in Table
2. Since the uncertainty to determine a grounding event is about $\pm 23$m, if some grid points of the
MIT have elevation difference $E_{dif}$ less than 23 m, we can conclude that this section of the
tongue is almost grounded. The smaller the $E_{dif}$, the more robust the grounding. From the color-
change patterns of Fig. 6a-d, we can see that part of the ice front grounded on the shallow Mertz
Bank from the end of 2002.
As illustrated from Table 2, the minimum$E_{dif}$ inside of the MIT are all less than 23 m
and the mean and minimum of the $E_{dif}$ in the buffer region are all less than 0 from 2002 to 2008.
From this, we conclude that the ice tongue has grounded on the shallow Mertz Bank since
November 14, 2002. This result coincides with findings from Massom et al. (2015) who
considered that the northwestern extremity of the MIT started to contact with the seafloor shoal
in late 2002 to early 2003. Also, it would be difficult for the MIT to approach the buffer region
(indicated with yellow to red color in Fig. 6) as the surrounding Mertz Bank gets shallower and
steeper, suggesting substantive grounding potentials. Inside of the MIT, the minimum of
elevation difference was just 11.9 m on November 14, 2002, which indicates little to no
grounding. However on March 8, 2004, December 27, 2006, and January 31, 2008, the
minimum of elevation difference reached -46.0 m, -52.3 m and -34.8m respectively, which
means significant grounding occurred in some regions. From 2002 to 2008, more regions under
the MIT have $E_{dif}$ less than 46 m, the area of which increased from 8 km$^2$ to 17 km$^2$.
Additionally, the mean of $E_{dif}$ under of the tongue for those having $E_{dif}$ less than 46 m
gradually decreases from 28.8 m to 12.3m, according to which we can conclude that the ice front
was grounded more significantly as time passed on. Additionally, since the grounding area
increased from 8 km$^2$ to 17 km$^2$ (Table 2) and the mean of $E_{dif}$ decreased from 2002 to 2008, we
can say that over the period from 2002 to 2008, the grounding of the northwest flank of the MIT
became more widespread.
Based on the calculated elevation difference, the grounding outlines of the MIT are
delineated for November 14, 2002, March 8, 2004, December 27, 2006 and January 31, 2008,
(Fig. 7). For the grounding part of the outline in different years, starting and ending location and
perimeter are also extracted, from which we can conclude that the length of the grounding
outline of the Mertz Bank is only limited to a few kilometers (Table 3).
We find that the lower right (northwest) of the MIT was always grounded and that
grounding did not occur in other regions (Fig. 6). The shallowest seafloor elevation the ice front
touched was ~ -290 m in November 2002.  In 2004, 2006, and 2008, the lower right (northwest)
of the MIT even approached the contour of –220 m.  Fig. 7 also shows the extension line of west
flank in November, 2002, from which we can see that if the ice tongue moved along the former
direction, the ice flow would be seriously blocked when approaching the Mertz Bank. The
shallowest region of the Mertz Bank has an elevation of about -140 m and the MIT would have
needed to climb the 140 m obstacle to cross it. The shallow Mertz Bank would have caused
grounding during the climbing. This special feature of seafloor shoal facing the MIT can further
explain why the ice velocity differed along the east and west flanks of the MIT before calving
and why the ice tongue moved clockwise to the east, as pointed out by Massom et al. (2015).
However, because of sparsely-distributed bathymetry data (point measurements) in Mertz region
used in Massom et al. (2015), this effect could not be easily seen. Here, from our grounding
detection results and surrounding high-accuracy bathymetry data, this effect is more clearly
observed.
**6. Discussion**

## 6.1 Area Changing Rate and ~70-year Calving Cycle of MIT

Using Landsat TM/ETM+ images from 1989 to 2013, outlines of the MIT are extracted manually. Assuming a fixed grounding line position over this period, the area of the MIT over this period is calculated. Using these data, from 1989 to 2007, an increasing area rate of the MIT is shown (from 5453 $km^2$ to 6126 $km^2$) in Fig. 8. However, the area of the MIT was almost constant from 2007 to 2010, before calving. The largest area of the MIT was 6113 $km^2$ closest to the calving event in 2010. After the calving, the area decreased to 3617 $km^2$ in November 2010.

The rate of area change for the MIT from 1989 to 2007 is also obtained using a least-squares method, corresponding to 35.3 $km^2$/a. However, after the calving a slight higher area-increasing trend of 36.9 $km^2$/a, is found (Fig. 8). On average, the area-increasing rate of the MIT was 36 $km^2$/a.

The surface behavior such as ice flow direction changes and middle rift changes caused by grounding was analyzed by Massom et al. (2015). In the history of the MIT, one or two large calving events were suspected to have happened between 1912 and 1956 (Frezzotti et al., 1998) and we consider it likely to be only once because of the influence of the shallow Mertz Bank. When the ice tongue touched the bank, the bank started to affect the stability of the tongue by bending the ice tongue clockwise to the east, as can be seen from velocity changes from Massom et al. (2015). With continuous momentum and flux input from upstream, a large rift from the west flank of the tongue would ultimately have to occur and could potentially calve the tongue. A sudden length shortening of the tongue can be caused by such ice tongue calving as indeed had happened in February, 2010. We also consider that even without a sudden collision of iceberg B-9B in 2010, the ice tongue would eventually calve because of existence of the shallow Mertz Bank.

If we take 6127 km$^2$ as the maximum area of the MIT, assuming a constant area-changing
rate of about 36.9 km$^2$/a after 2010, it will take about 68 years to calve again.  When assuming an
area changing rate of about 35.3 km$^2$/a as before 2010, it will take a little longer, about 71 years.
Therefore, without considering accidental event such as collision with other large icebergs, the
MIT is predicted to calve again in ~70 years.  Because of the continuous ice flow upstream, the
special location and relatively lower depth of the Mertz Bank, the calving is likely repeatable and
a cycle therefore exists.
After the MIT calved in February, 2010, Mertz polynya size, sea-ice production, sea-ice
coverage and high-salinity shelf water formation changed.  A sea-ice production decrease of
about 14-20% was found by Tamura et al. (2012) using satellite data and high-salinity shelf
water export was reported to reduce up to 23% using a state-of-the-art ice-ocean model
(Kusahara et al. 2010). Recently, Campagne et al. (2015) pointed out a ~70-year cycle of surface
ocean condition and high-salinity shelf water production around Mertz through analyzing
reconstructed sea ice and ocean data over the last 250 years. They also mentioned that this cycle
was closely related to presence and activity of Mertz polynya. However, the reason for this cycle
was not fully understood.
From these findings addressed above and MIT calving cycle we found, our explanation is
that the calving cycle of the MIT leads to the ~70-year cycle of surface ocean condition and
high-salinity shelf water production around Mertz. Calving decreases the length of the MIT
suddenly. Then, a short ice tongue reduces the size of Mertz Polynya formed by Antarctic
katabatic winds, resulting in lower sea-ice production and further lessens high-salinity shelf
water production. Therefore, the cycle of ocean conditions around Mertz found by Campagne et
al. (2015) is likely dominated by the calving of the MIT. Additionally, the 70 year cycles of MIT
calving coincides with surface ocean condition change around Mertz wellwhich makes the
explanation much more compelling.

**6.2 Key issues influencing grounding detection**

Several issues on grounding detection require further clarification, such as sea surface
height, FAC value and accuracy of seafloor DEM. In this section, their influences on final
grounding detection results are more deeply discussed.

**6.2.1 The Lowest Sea-Level Extraction**

In Section 3.1, the lowest sea level -3.35 m is derived by comparing all sea-surface
heights derived from different tracks and campaigns from 2003 to 2009. This constant stands for
the lowest sea level from results around Mertz from 2003 to 2009 and is directly from
ICESat/GLAS observation. However, because of limited observations in each year,
ICESat/GLAS may not catch the lowest one. Sea level lower than -3.35 m may exist over Mertz
region which would make the grounding results more severe with occurance of more negative
values in Fig. 6.

**6.2.2 Firn Air Content Calculation**

FAC varies across the Antarctica ice sheet, usually decreasing from the interior to the
coast. In Section 3.2, FAC over Mertz region is derived as $4.87\pm1.31$ m. However other time
dependent modeling results from the Mertz region were closed to 5-10 meters (Ligtenberg et al.
2014). Since there are no in-situ measurements available for verification, further comparison
work needs to be conducted. However, this FAC value is derived according to our best
knowledge over Mertz and is affected by iceberg status (using our approach) and the maximum
freeboard used.

First, for FAC calculation, icebergs just touching the seafloor should be used in which

case the FAC calculated assuming hydrostatic equilibrium is the same as the actual value.
However, it is difficult to ascertain whether an iceberg is just touching the seafloor from remote
sensing images. The near stationary or slowly rotating iceberg detected with remote sensing
should be grounded more severely than just touching the seafloor, which may result in a
calculated FAC theoretically larger than the actual value. Thus, using this FAC result to detect
grounding can potentially lead to smaller grounding results. However, once an iceberg or ice
tongue is detected as grounded, the result is more convincing.

Second, because ICESat/GLAS observed only several times a year on repeat tracks and

icebergs was rotating slowly, the elevation profile in 2006 and 2008 along the same track T1289
may not come from the same ground surface.  Fig. 9 shows the freeboard over iceberg 'A', 'B'
and 'C' derived from ICESat/GLAS from 2006 and 2008. By comparing freeboard of iceberg 'A'
in 2006 (Fig. 9a), and 2008 (Fig. 9c), we can find that the maximum freeboard was larger and the
freeboard profile was longer in 2006. Comparatively, the smaller freeboard in 2008 may be
caused by ice basal melting or observing different portion of iceberg 'A'. Since the larger
freeboard measured in 2006 indicates a high possibility of capturing the thickest portion, the
freeboard measurement in 2006 is used to invert the FAC.  Additionally, iceberg 'A' and 'C' did
show the similar maximum freeboard (Table 1), which is another important reason to select the
measurement in 2006 to invert.
**6.2.3 Seafloor DEM**

High accuracy seafloor elevation is critical to the final success of grounding detection. As

can be seen from S-Fig.1, there is no bathymetry data under the MIT, which may result in large
uncertainty for seafloor interpolation. The oldest bathymetry data collected along the margin of
the MIT was at least from 2000 (Beaman et al. 2011). Thus, the boundary of the MIT in 2000 is
used to identify bathymetry measurement gaps, as is indicated in Fig. 6. But around the Mertz ice
front, for both the east and west flanks, bathymetry data does exist, which provides control points
for seafloor interpolation under the tongue. Since the ice front has a width of ~34 km (Wang et al.
2014), the accuracy of seafloor DEM under the MIT varies according to different distances to the
control points. Inside of the MIT boundary of 2000, the closer to the dash-dotted polygon (Figs.
6 and 7), the better the accuracy the seafloor DEM. Outside of that boundary, the quality of the
seafloor DEM data is much better because of the high density of single-beam or multi-beam
bathymetry measurements.

However, from Beaman et al. (2011), no uncertainty on the seafloor DEM was

systematically provided. Instead, only the poorest accuracy of single or multi-beam bathymetric
measurements was available. Since no new bathymetry data is publicly available in this region, it
is not possible to conduct further work on evaluation of the seafloor bathymetry and interpolation
error from kriging using bathymetry data is difficult to assess.  Thus, the accuracy under poorest
situation for bathymetry data is used, the same as used in Beaman et al. (2011).

Since Beaman et al. (2011) provided the most accurate seafloor DEM over Mertz

according to our best knowledge, seafloor DEM inside of dash-dotted polygon (Fig. 7) is kept
and the grounding detection is conducted there (Fig. 6) as well. Additionally, the ice tongue
never stopped flowing further into the ocean, where the bathymetry measurements density is
good.  From results shown in Fig. 6 all grounding sections of MIT boundary are located outside
of the 2000 boundary. Thus the analysis of grounding detection near ice front in 2002, 2004,
2006, and 2008 is convincing. Inside of the 2000 boundary, most of the grounding detection
results are above 100 m, indicating a floating status of the corresponding ice. Only abnormal
seafloor features higher than this seafloor DEM by about 100 m can result in wide grounding
inside.  Additionally, from surface features of the MIT from Landsat TM/ETM+ images, no
abrupt sunlight shadow related to grounding is detected from 1989 to 2010 near the front, which
indicates that the judgment of floating ice tongue inside of the 2000 boundary from Fig. 6 is
correct.  Actually, no matter whether the MIT inside of the 2000 boundary was grounded or not,
gradual grounding on the shallow Mertz Bank of the MIT since late 2002 is a fact, which is
direct evidence for us to infer the primary cause of the instability of the MIT.
**7. Conclusion**

In this study, a method of FAC calculation from seafloor-touching icebergs around Mertz

region is presented as an important element of understanding MIT grounding. The FAC around
the Mertz is about $4.87\pm1.31$ m. This FAC is used to calculate ice draft based on sea level and
freeboard extracted from ICESat/GLAS and appears to work well. A method to extract
grounding sections of the MIT is described based on comparing inverted ice draft assuming
hydrostatic equilibrium with seafloor bathymetry. The final grounding results explain the surface
behavior of the MIT. Previous work by Massom et al. (2015) has also provided some evidence
for seafloor interaction, in showing that the MIT front had an approximate 280 m draft with the
nearby seafloor as shallow as 285 m, suggesting the possibility of grounding.  In our work, we
have provided ample detailed bathymetry and ice draft calculations.  Specifically, ice bottom
elevation is inverted using ICESat/GLAS data and compared with seafloor bathymetry during
2002, 2004, 2006, and 2008. From those calculations we show conclusively that the MIT was
indeed grounded along a specific portion of its northwest flank over a limited region. We also
point out that even without collision by iceberg B-9B in early 2010 the ice tongue would
eventually have calved because of momentum and flux input from the upstream glacier flow
being increasingly opposed by a reaction force from the shoal of the Mertz Bank.
From remote sensing images we are able to quantify the rate of increase of area of the
MIT before and after the 2010 calving.  While the area-increasing trend of the MIT after calving
is slightly larger than before, we use the averaged rate to estimate a timescale required for the
MIT to re-advance to the area of the shoaling bathymetry from its retreated, calved position.  Our
estimate is ~70-years, which is remarkably consistent with Campagne et al. (2015) who found a
similar period of sea surface changes using seafloor sediment data. A novel point we bring out in
our study is that it is the shoaling of the seafloor combined with the rate of advance of the MIT
that leads to the 70-year repeat cycle.  Also the calving cycle of the MIT explains the observed
cycle of sea surface conditions change well, which indicates the calving of the MIT is the
dominant factor for sea-surface condition change. Understanding the mechanism underlying the
periodicity of MIT calving is important as the presence or absence of the MIT has a profound
impact on sea ice and hence of bottom water formation in the local region.
**Acknowledgements**
This research was supported by Fundamental Research Fund for the Central University,
the Center for Global Sea Level Change (CSLC) of NYU Abu Dhabi (Grant: G1204), the Open
Fund of State Key Laboratory of Remote Sensing Science (Grant: OFSLRSS201414),  and the
China Postdoctoral Science Foundation (Grant: 2012M520185, 2013T60077). We are grateful to
the Chinese Arctic and Antarctic Administration, the European Space Agency for free data
supply under project C1F.18243, the National Snow and Ice Data Center (NSIDC) for the
availability of the ICESat/GLAS data (http://nsidc.org/data/order/icesat-glas-subsetter) and
MODIS     image     archive     over     the     Mertz     glacier     (http://nsidc.org/cgi-
bin/modis_iceshelf_archive.pl), British Antarctica Survey for providing Bedmap-2 seafloor
topography data (https://secure.antarctica.ac.uk/data/bedmap2/), the National Geospatial-
Intelligence Agency for publicly released EGM2008 GIS data (http://earth-
info.nga.mil/GandG/wgs84/gravitymod/egm2008/egm08_gis.html), and the USGS for Landsat
data (http://glovis.usgs.gov/). Fruitful discussions with M. Depoorter, P. Morin, T. Scambos and
R. Warner, and constructive suggestions from Editor Andreas Vieli and two anonymous
reviewers are acknowledged.

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

**Figures**

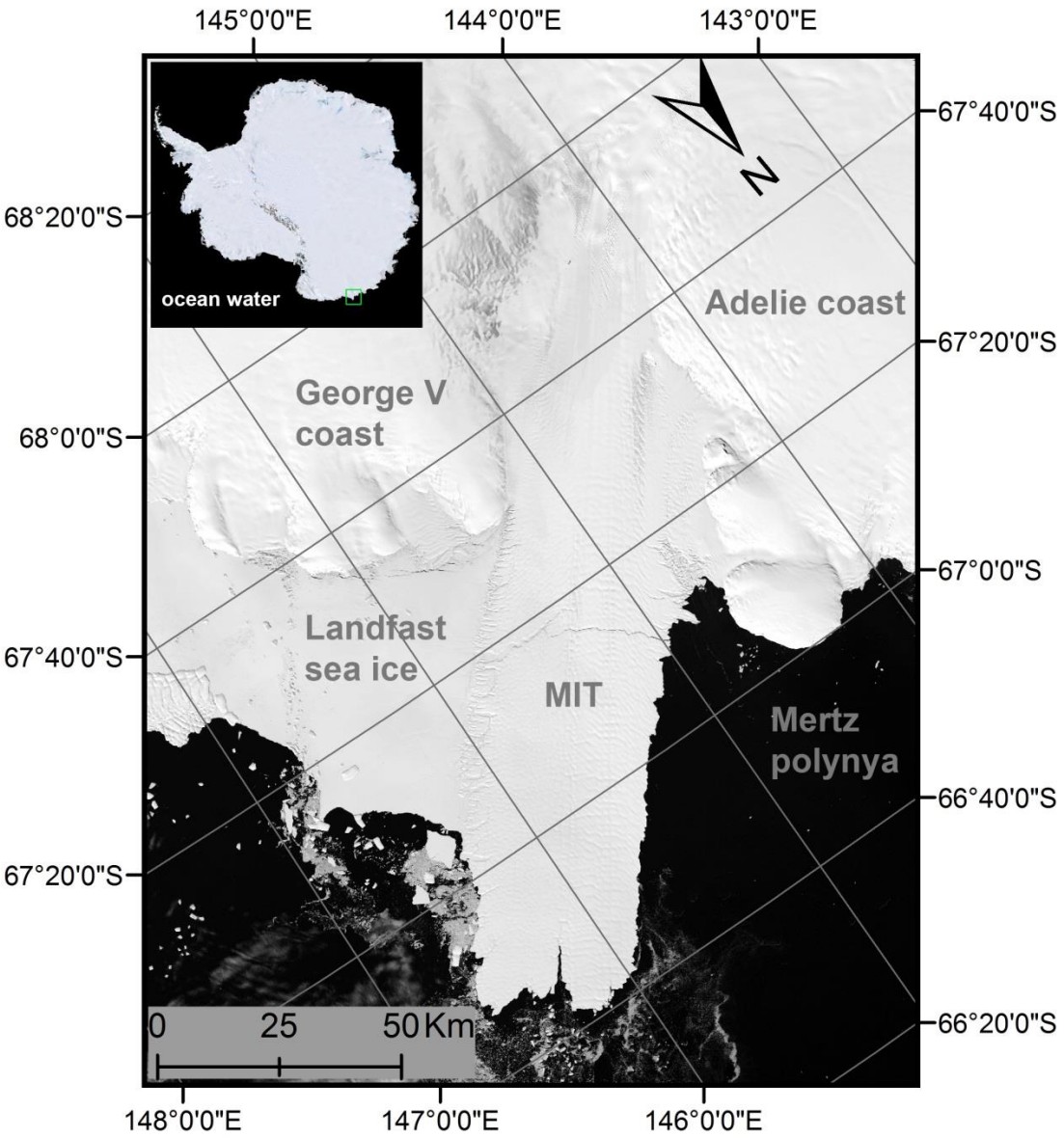


**Figure 1.** Mertz Ice Tongue (MIT), East Antarctica. Landfast sea ice is attached to the east flank
of the MIT and the Mertz Polynya is to the west. The background image is from band 4 Landsat
7, captured on February 2, 2003. The green square found in the upper left inset indicates the
location of the MIT in East Antarctica. A polar stereographic projection with -71°S as standard
latitude is used.

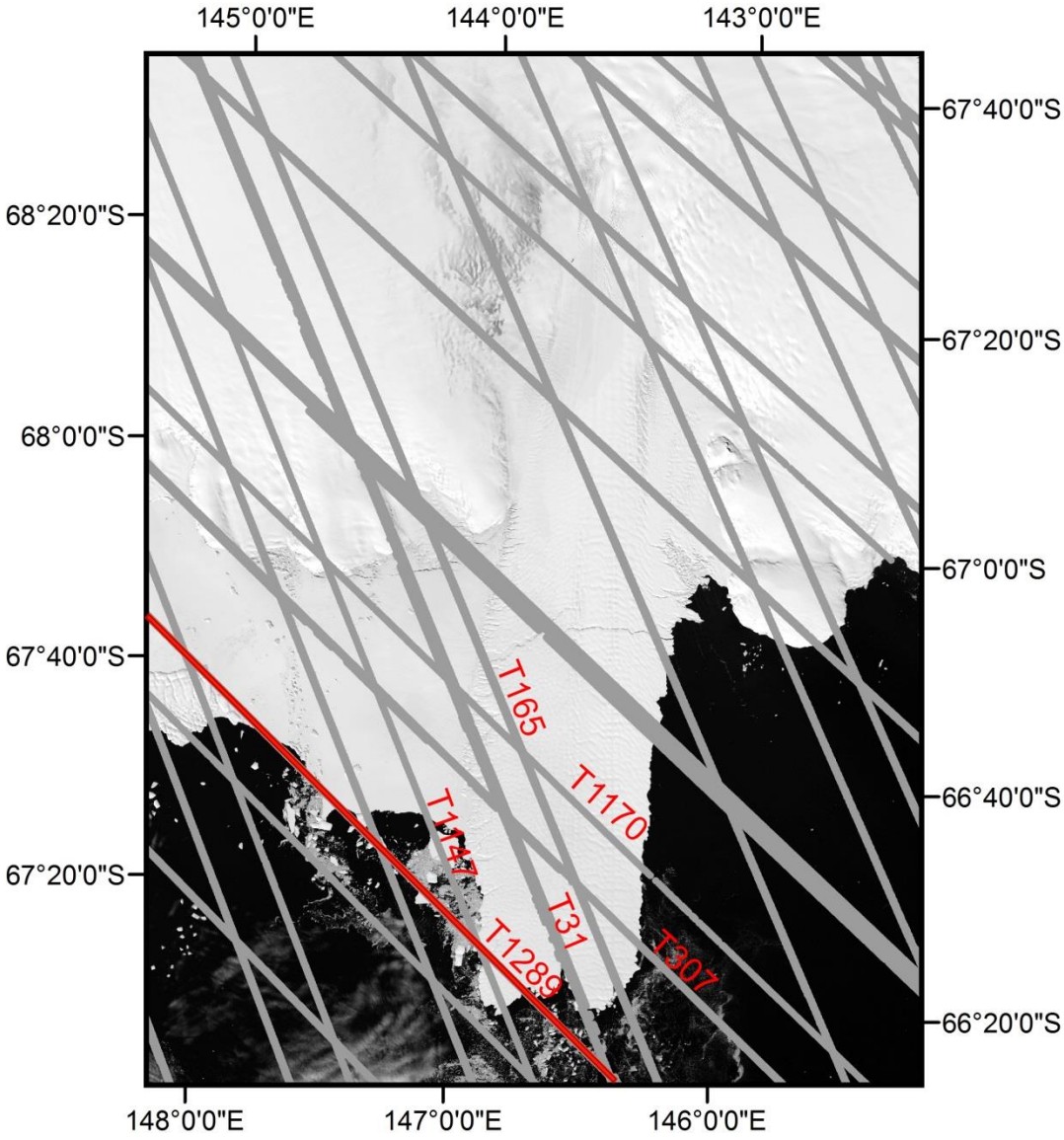


**Figure 2.** Spatial distribution of ICESat/GLAS data from 2003 to 2009 covering the Mertz

region. Ground tracks of ICESat/GLAS are indicated with gray lines. Track 1289 (T1289) is

highlighted in red as is used in Fig. 4. The background image is from band 4 Landsat 7, captured

on February 2, 2003. A polar stereographic projection with -71 ͦS as standard latitude is used.

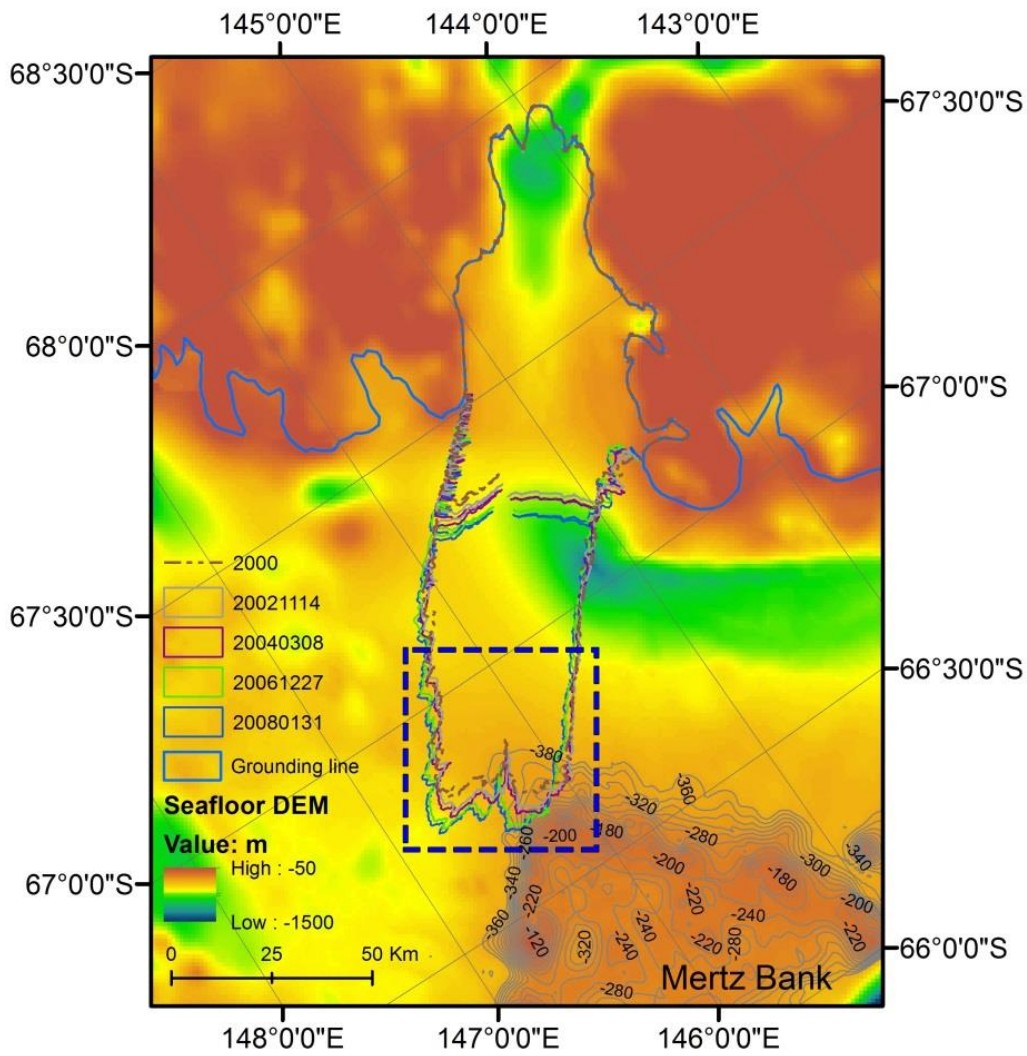

**Figure 3**. Seafloor topography from bathymetry around Mertz region and outlines of the MIT

from 2002 to 2008. The outlines of the MIT in different years are marked with different colored

polygons. The shallow Mertz Bank is located in the lower right (northeast). The dash-dotted line

indicates the shape of the MIT on January 25, 2000, which is used to identify the bathymetry gap

under the ice tongue. The dash-dotted blue inset box corresponds to location of Fig, 6 and 7. The

bathymetry measurement profile can be found from S-Fig. 1.

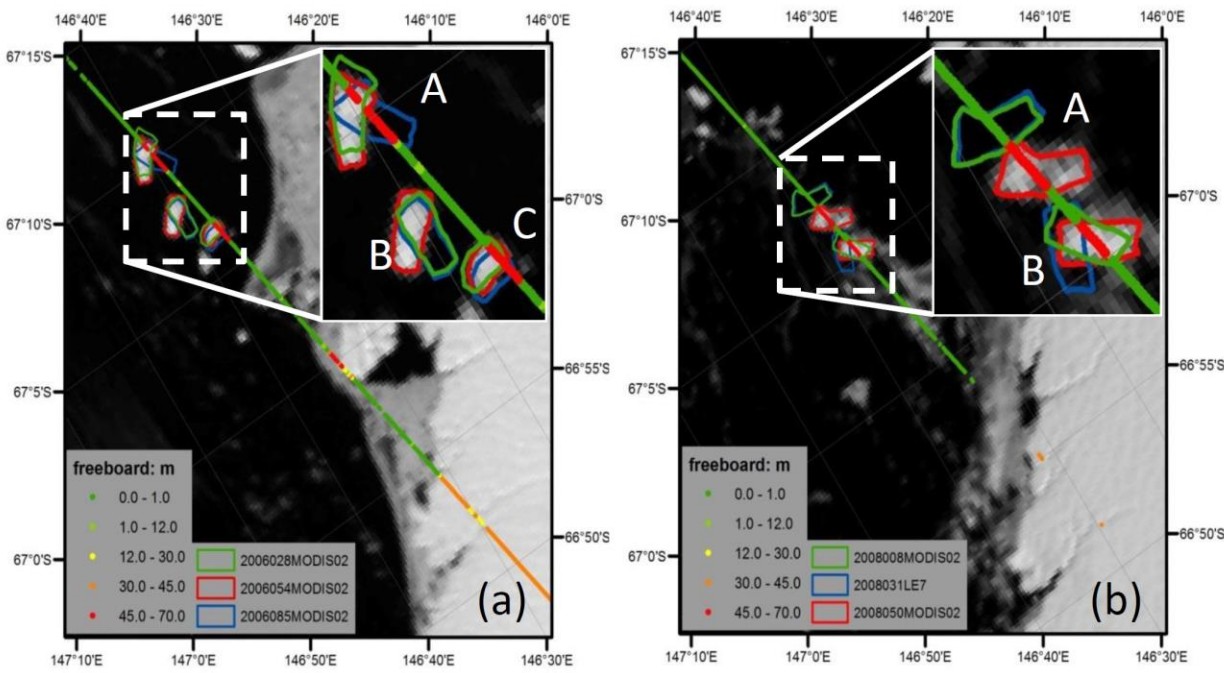

**Figure 4.** Freeboard extracted from Track 1289, ICESat/GLAS, the location of which can be

found in Fig. 2 and S-Fig. 1. (a) and (b) show the freeboard extracted from ICESat/GLAS on

February 23, 2006 (2006054 ) and February 18, 2008 (2008049) respectively. In each image,

positions of three icebergs (with name labeled as 'A', 'B' and 'C') closest to ICESat/GLAS

observation time are plotted with green, red and blue polygons respectively. The dates are

indicated with seven numbers (yyyyddd) in legend. 'yyyyddd' stands for day 'ddd' in year

'yyyy'. 'MODIS02' and 'LE7' indicate  that  the image used to extract  iceberg outline is from

MODIS and Landsat 7 ETM+, respectively.


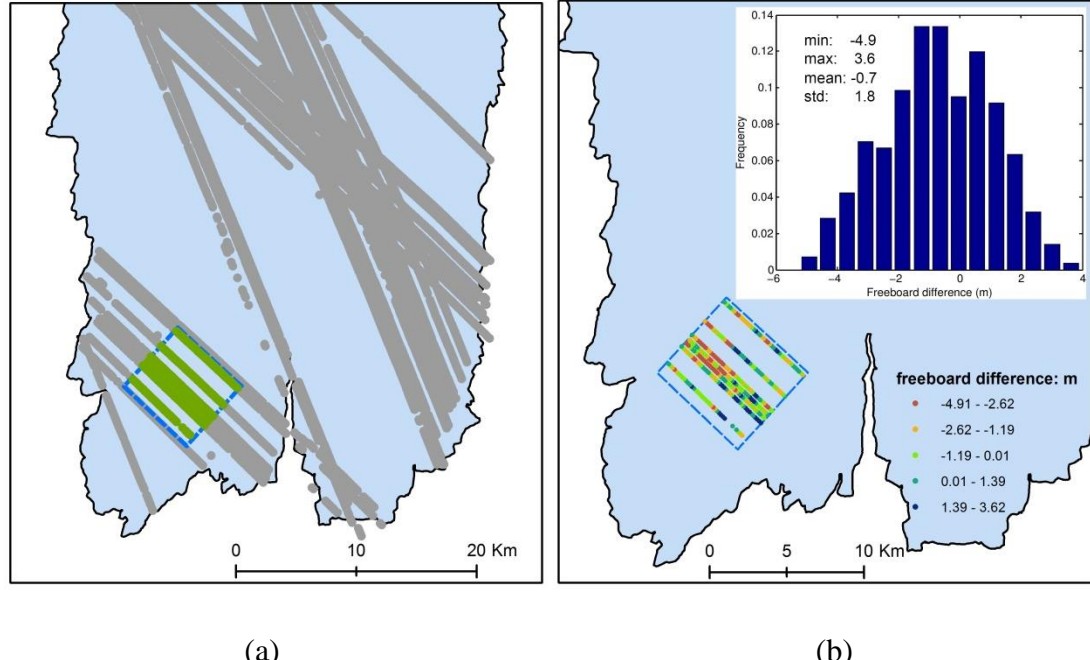


(a)                                   (b)

**Figure 5.** Evaluation of kriging interpolation method over the MIT using freeboard data derived
from ICESat/GLAS. (a) shows profile locations of freeboard derived from ICESat/GLAS after
relocation over the MIT. Gray dots indicate ICESat/GLAS used for interpolation using kriging
method. The blue dashed square indicates the region used to investigate interpolation accuracy of
kriging method, about 7 km×7 km. Inside of the square, freeboard data marked with green dots
are used to check the accuracy of freeboard interpolated with kriging. (b) is the freeboard
comparison result derived by subtracting krigged freeboard from freeboard derived from
ICESat/GLAS. The spatial distribution and the histogram of freeboard difference are shown in
the lower left and upper right respectively. The black polygon filled with light blue shows the
boundary of MIT on November 14, 2002.



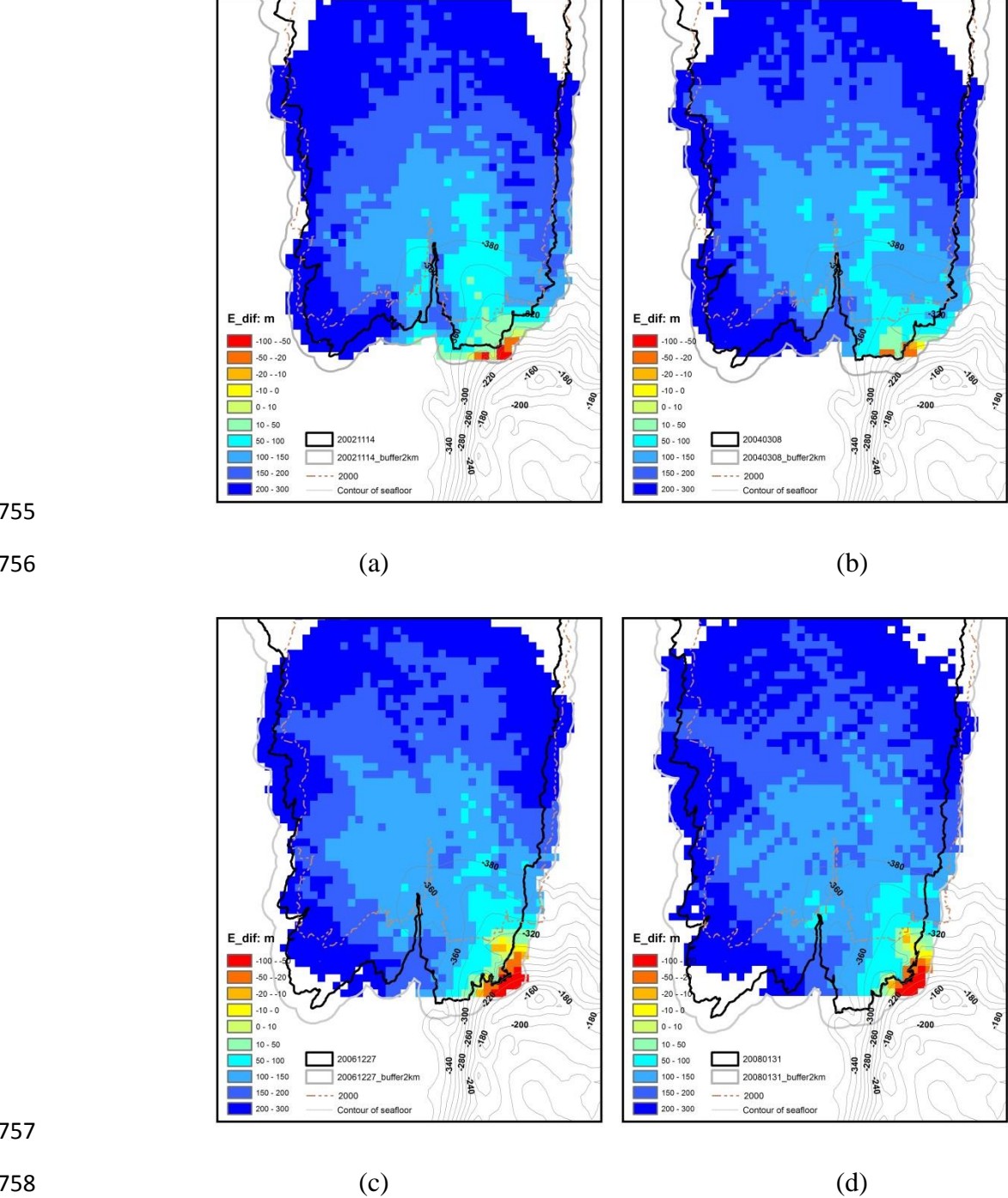


(a)                                               (b)


(c)                                               (d)

**Figure 6.** Elevation difference of Mertz ice bottom and seafloor topography.  (a), (b), (c) and (d)

correspond to elevation difference assuming hydrostatic equilibrium under the minimum sea

surface height -3.35 m on November 14, 2002 , March 8, 2004, December 27, 2006, and January

31, 2008, respectively. The contours in the lower right indicate seafloor topography (unit: m) of
the Mertz Bank with an interval of 20 m.  The solid black line indicates the boundary of the MIT
and the thick gray line outlines a buffer region of the boundary with 2 km as buffer radius. The
dash-dotted line indicates the shape of the MIT on January 25, 2000, which is used to identify
the bathymetry gap under the ice tongue. In the legend, negative values mean that ice bottom is
lower than the seafloor, which of course is impossible. Therefore, the initial assumption of a
floating ice tongue was incorrect in those locations (yellow to red colors), and the ice was
grounded. Regions with more negative values indicate more heavily grounding inside of the MIT
or more heavily grounding potential in the buffer region.

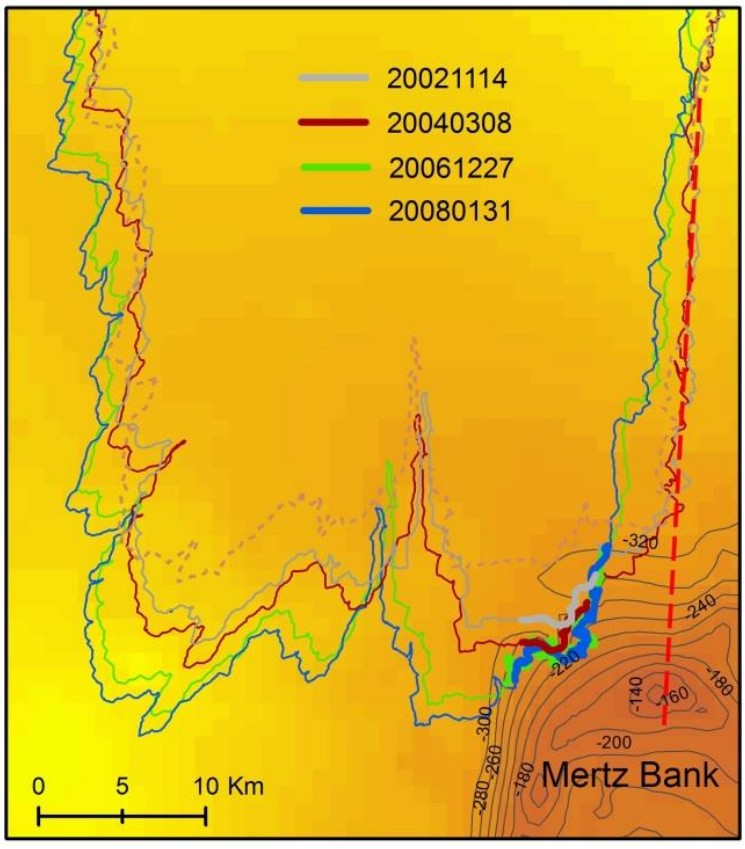


**Figure 7**. Digital Elevation Map (DEM) of seafloor around Mertz and grounding section of the

boundaries extracted from 2002 to 2008. The grounding sections of the MIT boundary in 2002,

2004, 2006 and 2008 is marked with thick gray, purple, green and blue polylines respectively

and MIT boundaries are indicated with polygons with the same legend as Fig. 3. Additionally,

MIT boundary in 2000 indicated with dash-dotted polygon is used to show the different quality

of seafloor DEM. Inside of this polygon no bathymetry data was collected or used. The dashed

red line indicates the 'extension line' of the west flank of MIT on November 14, 2002, passing

the shallowest region of the Mertz Bank (about -140 m).

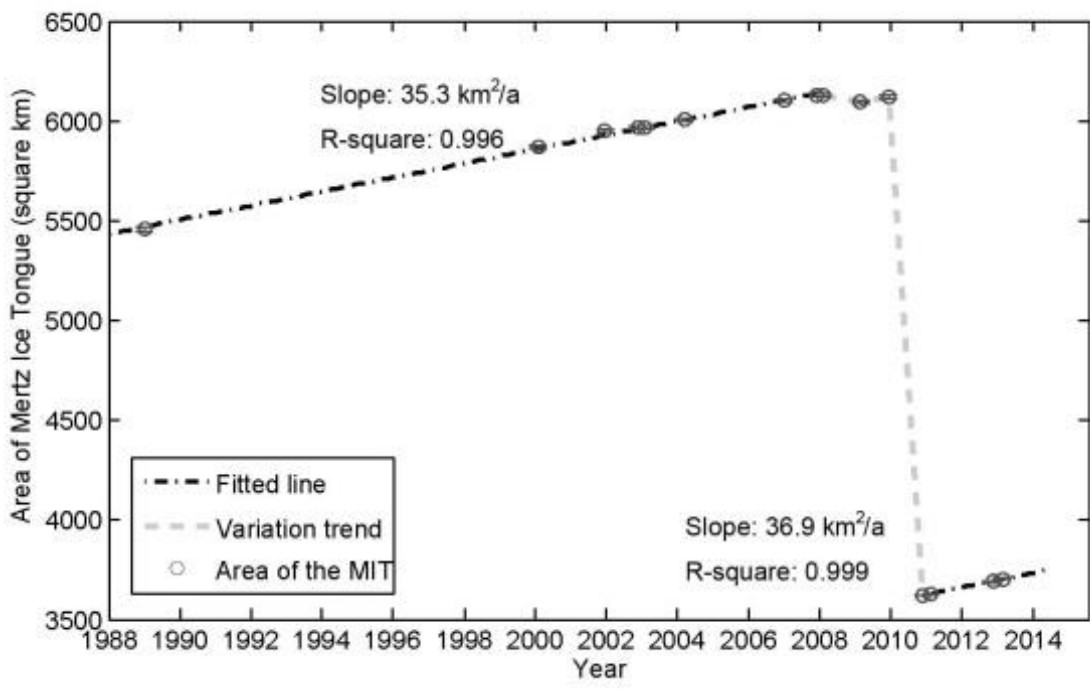


**Figure 8**. Time series of area change of the MIT. The area covers the entire ice tongue, to the

grounding line as indicated with thick blue line in Fig. 3. The area is extracted from Landsat

images from 1988 to 2013.



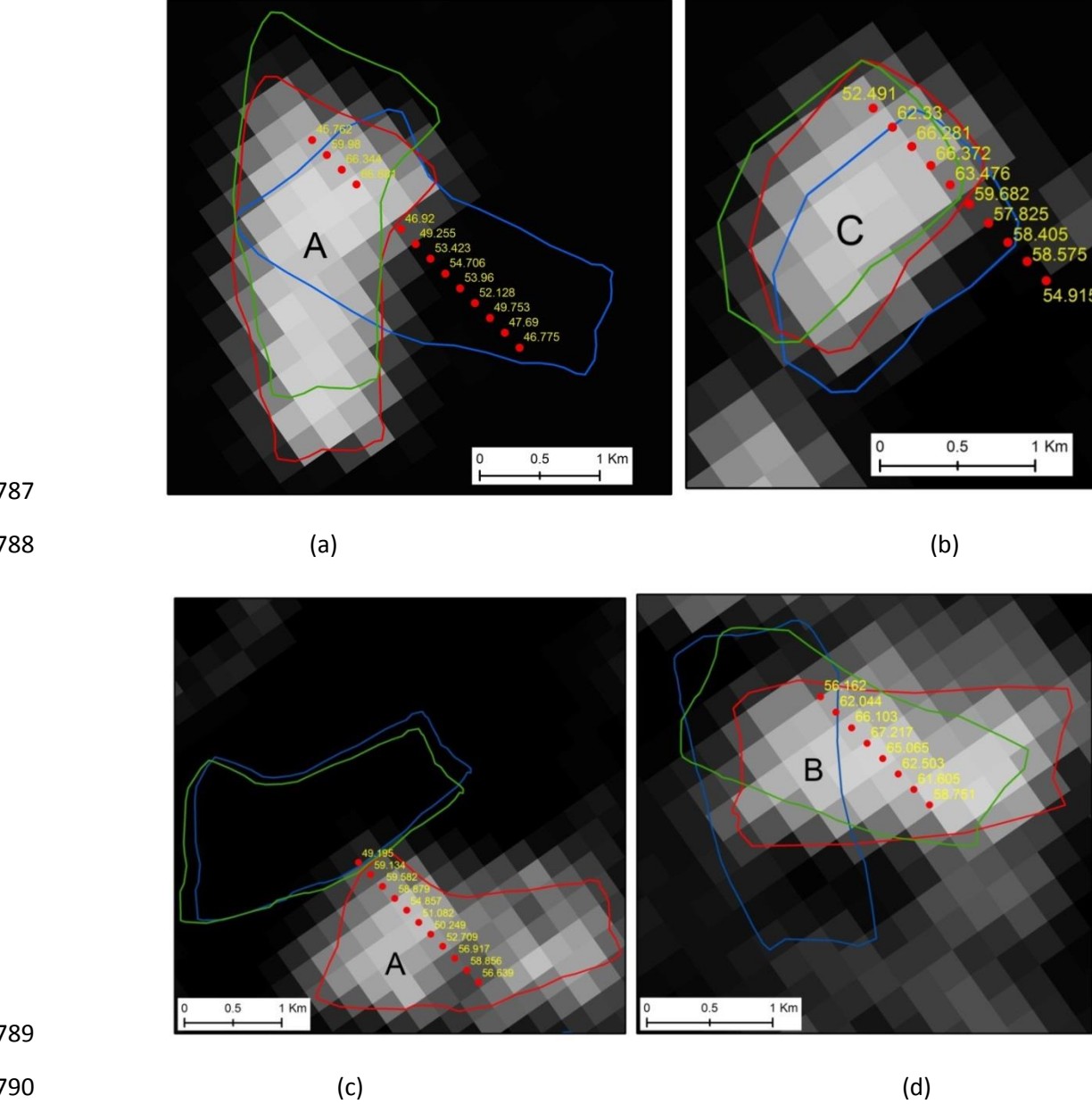


(a)                                    (b)


(c)                                    (d)

**Figure 9**. Freeboard extraction results from ICESat/GLAS for icebergs 'A', 'B' and 'C' in 2006
and 2008 respectively. (a) and (b) correspond to freeboard measurements from 'A' and 'C'
respectively on February 23, 2006 (2006054), with background image from MODIS captured on
2006054. (c) and (d) correspond to freeboard measurements  from 'A' and 'B' respectively on
February 18, 2008 (2008049), with background image from MODIS captured on 2008050. The
location of each iceberg in different observation time is indicated with different colored polygons,
the legend of which is the same as what is used in Fig. 4. Inside of each sub-figure, different
icebergs are marked with capital characters 'A', 'B' and 'C' respectively and iceberg freeboard
results in unit of meter are marked in yellow.

**Tables**
**Table 1**. Statistics of the three icebergs used to inverse FAC with least-square method. Icebergs
'A', 'B' and 'C' are the same as what are used in Fig. 4 and 9. Measurements from icebergs 'A'
and 'C' in February 2006 are used to derive FAC with least-squares method. Icebergs 'A' and 'B'
in 2008 are used for validation.

| Icebergs | date | Latitude ($°$) | Longitude ($°$) | Freeboard (m) | Seafloor (m) | Sea level (m) | $\varepsilon$ (m) | $E_{dif}$ (m) |
|----------|------|----------|-----------|-----------|----------|-----------|-------|-------|
| A | Feb 23, 2006 | -67.1737 | 146.6595 | 66.88 | -528.48 | -1.92 | 0.89 | |
|   |      | -67.1752 | 146.6604 | 66.34 | -527.01 | -1.92 | 1.30 | |
| C | Feb 23, 2006 | -67.1085 | 146.6247 | 66.37 | -505.84 | -1.92 | -1.25 | |
|   |      | -67.1100 | 146.6255 | 66.28 | -507.08 | -1.92 | -1.01 | |
| A | Feb 18, 2008 | -67.1194 | 146.6303 | 58.88 | -522.52 | -2.08 | | 69.14 |
|   |      | -67.1209 | 146.6311 | 59.58 | -524.16 | -2.08 | | 64.88 |
| B | Feb 18, 2008 | -67.0906 | 146.6151 | 67.22 | -500.92 | -2.08 | | -22.45 |
|   |      | -67.0921 | 146.6159 | 66.10 | -500.47 | -2.08 | | -13.55 |


**Table 2**. Statistics of grounding grids inside or grounding potentials outside of the Mertz Ice
Tongue (MIT) ( 'I': inside of thick black line, Fig. 6; Number in brackets indicates how many
grids are located inside of the 2000 Mertz boundary;  'O': between the black and gray lines, Fig.
6) on November 14, 2002, March 8, 2004, December 27, 2006 and January 31, 2008 respectively.
Each grid covers an area of 1 km$^2$. The Mean, Minimum and Standard deviation is calculated
without considering those fallen inside of the 2000 Mertz boundary, but only those having
elevation difference less than 46 m and out of 2000 Mertz boundary.

| Elevation difference (subtracting seafloor from ice bottom) | 2002-11-14 | | 2004-03-08 | | 2006-12-27 | | 2008-01-31 | |
|---|---|---|---|---|---|---|---|---|
| | I | O | I | O | I | O | I | O |
| 23-46 (m) | 9(3) | 10(0) | 6(0) | 3(0) | 10(1) | 1(0) | 10(3) | 5(0) |
| 0-23 (m) | 2(0) | 6(0) | 1(0) | 1(0) | 9(0) | 2(0) | 4(0) | 2(0) |
| <0 (m) | 0(0) | 8(0) | 2(0) | 5(0) | 7(0) | 21(0) | 6(0) | 18(0) |
| Mean (m) | 28.8 | 9.8 | 15.8 | -1.1 | 10.9 | -41.9 | 12.3 | -31.0 |
| Minimum (m) | 11.9 | -81.5 | -46.0 | -44.5 | -52.3 | -102.8 | -34.8 | -103.0 |
| Standard deviation (m) | 9.2 | 36.8 | 29.6 | 31.4 | 24.7 | 37.6 | 27.3 | 38.0 |
| Number of grids | 8 | 24 | 9 | 9 | 25 | 24 | 17 | 25 |


**Table 3**. Statistics of grounding outlines of the MIT as shown with thick polylines in Fig. 7 on
November 14, 2002, March 8, 2004, December 27, 2006 and January 31, 2008 respectively

|  | 2002-11-14 | 2004-03-08 | 2006-12-27 | 2008-01-31 |
|---|---|---|---|---|
| Start location (°) | 146.124 °E, 66.696 °S | 146.155 °E, 66.681 °S | 146.093 °E, 66.700 °S | 146.088 °E, 66.699 °S |
| End location (°) | 146.240 °E, 66.693 °S | 146.256 °E, 66.683 °S | 146.304 °E, 66.669 °S | 146.292 °E, 66.668 °S |
| Perimeter (km) | 7.0 | 6.4 | 24.7 | 20.9 |
