# Peer review of "Grounding and Calving Cycle of Mertz Ice Tongue"

_The Cryosphere, 2016_

## Referee Comment (RC1) · Anonymous Referee #1 · 15 Feb 2016

This paper uses altimetry and bathymetry data to map changes in the grounded area of the Mertz ice tongue during the early-to-mid 2000s. They use the surface heights of lightly-grounded icebergs to estimate the firn air content for the ice tongue, and use this, geoid-corrected altimetry measurements, and a bathymetry map of the ice shelf, to map the difference between the bottom of the ice shelf, at hydrostatic equilibrium, and the sea bed for different time periods. Areas where these maps show the (hydrostatic) ice bottom below the seabed are treated as grounded. The authors find that the northwest flank of MIT (Mertz Ice Tongue) was grounded during 2002-08, and that the grounding increased between 11/2004 and 12/2006. They propose that the MIT would have calved because of increasing grounding extent even if the tongue had not been hit by an iceberg. The authors also examine the rate of change in the area of MIT, and estimate an interval between subsequent tongue calving events of 70 years.

A quick summary of this review is that this manuscript needs a great deal of editing and extensive revision before it is ready for publication. I have presented some of my thoughts about what needs to be fixed, but the writing is of uneven quality and my comments about the reliability of interpolated data should (in my opinion) lead to substantial changes in the text and figures. With this in mind, I have not gone to the effort of editing the paper in detail, and hope that the authors can do so themselves.

A major problem in this manuscript is the lack of bathymetry data under the MIT. Data are scattered, at varying density, seaward of the ice front, and the bathymetry maps appear to resolve the seaward extent of the Mertz Bank, but under the tongue the maps are entirely based on interpolated values. This makes the maps in figure 5 and the statistics in table 2 hard to believe except at the very edge of MIT where the altimetry and bathymetry more or less coincide. The conclusions of the paper are largely independent of the data everywhere except in the area where the data are credible, which makes me wonder why the authors chose to show the mapped elevation-difference values in the areas for which they have no data. The authors should make a clear distinction between results derived from measurements and results derived from interpolated values, and the relative expected accuracies of each.

A second problem is that the methods are difficult to interpret, in large part because the report cited as "Wang 2014" is not readily available online, and the paper "Wang et al 2014" appears to describe a method for estimating freeboard change, rather than the absolute freeboard used in the present study. A good deal of the material in this paper is based on a technique in that report, described briefly in section 3.1(126-135). This paper should include a full description of the technique. In particular, it is not clear from the description how the relocation step works or what it is supposed to accomplish, or how the surface slope relates to errors in this relocation (line 241). I would also have liked to see a justification for the kriging interpolation between ICESAT profiles; the grounding features appear to be small compared to the gaps between ICESAT tracks, which makes me suspect that the krigged freeboard values may not provide a good

indication of grounding.

The authors should also be clear about the tidal values used in the freeboard study. Are the altimetry values corrected for tides? What is the "lowest sea level" mentioned at 155, and elsewhere? Is it derived from a tide model, or is it the lowest observed sea level? Is the tide model on the ICESAT product used, or is a different tide model used? What are the errors involved in each part of this?

The English in the manuscript needs improvement. A few idioms are used throughout that are confusing or distracting. "Inversed" should be "inverted." "Area-changing rate" should just be "area rate." "Least-square" should be "least-squares." Activities in the current study should be in present tense, citations to the literature should be in past tense.

The FAC calculation (3.2) has some nice features, but needs to be described in more detail. How is the least-squares inversion carried out? What are the error sources?

160-177- is the extensive discussion of other methods of calculating the FAC germain to this study? This section would be clearer if much of this were omitted.

222-229- this paragraph should be in the introductory part of section 3.2, not after the calculations have been presented.

247: Where are the interpolation errors for freeboard and bathymetry?

254- 50 times the average slope is still a very small number (0.6 degrees). A better estimate of the error due to crevassing would be to directly incorporate the crevasse depth into the calculation- thus, instead of v*slope_error ($\sim$ 12 m) the contribution would be closer to 50 m.

256-62: Why do we need to consider the freeboard stable (or not stable?) It appears that only static estimates of freeboard are used here (derived from single ICESAT campaigns) - so why does it matter that there would be a change (or not) in the freeboard?

257 "annual changing rate of freeboard" should be "annual rate of freeboard change" or "freeboard rate"

273: What is the significance of Edif < 34 m? Based on 263-268, this would indicate "not extremely confidently identified as ungrounded." Wouldn't a better statistic be Edif < -34, or "Extremely confidently identified as ungrounded?"

280: Again: Do you mean "less than -17 m?"

291-293: Reporting Edif within the tongue is a problem, since the bathymetry is not known there. You might report changes along the margin, but the statistics reported here don't seem to mean anything.

302 Combine the first two sentences, which form a joint conclusion: ". However," should be ", and that"

325 (and elsewhere) "Area-expanding trend" should be "area rate" or "rate of area change"

367-78: The significance of this paragraph is not clear. Ice-berg scouring is not discussed elsewhere in the paper, so the scientific question addressed by this paragraph needs more introduction.

577- "is used in figure 6"– this appears not to be true.

589: "closed' – should this be "closest?"

594: The legend here is not consistent with the caption.

606: It is hard to distinguish the outline from the "grounding part." The choice of colors (yellow on yellow) is not good.

---

## Referee Comment (RC2) · Anonymous Referee #2 · 22 Feb 2016

The authors use bathymetric, ICESat, and Landsat data products to estimate the firn air content, depth below sea level, re-grounding locations, and advance rate for the Mertz Ice Tongue from 2002-2008. They find that grounding along the Mertz Bank resulted in slight rotation and rifting of the Mertz Ice Tongue that would have resulted in the ice tongue's eventual collapse in the absence of any additional triggering mechanisms. Further, they suggest that the ice tongue collapse has a periodicity of ∼70 years and that this periodicity results in periodic variations in local sea ice formation and bottom water formation. Although the topic of the manuscript is interesting, the limited presentation of the methods and irregular quality of the writing make it difficult to follow. In addition to the major revisions listed below, I recommend that the authors go through the text in detail to check the writing and to make sure that all figures are legible.

1) In the data and methods sections, the authors frequently refer the readers to other publications rather than describe the data processing procedures in detail in the text. I find this to be particularly concerning for the freeboard inversions to estimate ice thickness because small errors in freeboard can lead to large variations in the estimated tongue depths. In order to have confidence in the provided tongue depths, I recommend including more detail on the relocation and interpolation procedures. Similarly, more information regarding the uncertainty of the bathymetry data used to identify grounded regions would be incredibly helpful.

2) It's really difficult to follow the firn air content approximation. I assume the bed elevations are really well constrained under the targeted icebergs and you are simply iteratively estimating the iceberg depths for gradually decreasing values of the mean iceberg density. The units obtained for the firn air content estimated using this method require explanation. I assume that they represent the difference in iceberg depth assuming a constant ice density and the final ice density estimated through the comparison with the underlying bathymetry since the units are in meters, but this is not presented anywhere. It would be helpful to also present the final density inferred for the firn column so that it is easier to compare your estimates with other observations. The error estimates obtained for firn air content should also be presented in more detail. I am particularly concerned with the assumption that the ICESat tracks capture the thickest portion of each iceberg. I'd be more confident in the firn air content estimates if I was also shown that there are relatively small variations in iceberg freeboard along the ICESat tracks because that would increase confidence that the iceberg grounding location is captured by the ICESat data.

3) The addition of the iceberg scour section at the end of the discussion is somewhat out of place with the rest of the manuscript. I suggest removing it entirely.

---

## Author Comment (AC1) · 6 Apr 2016

**We are grateful to the reviewers for their time and constructive comments to improve this manuscript. Here we address our reply point by point, in bold font. Reviewer's comments are in regular font. All changes are marked with red in the revised manuscript.**

Reviewer 1:

This paper uses altimetry and bathymetry data to map changes in the grounded area of the Mertz ice tongue during the early-to-mid 2000s. They use the surface heights of lightly-grounded icebergs to estimate the firn air content for the ice tongue, and use this, geoid-corrected altimetry measurements, and a bathymetry map of the ice shelf, to map the difference between the bottom of the ice shelf, at hydrostatic equilibrium, and the sea bed for different time periods. Areas where these maps show the (hydrostatic) ice bottom below the seabed are treated as grounded. The authors find that the northwest flank of MIT (Mertz Ice Tongue) was grounded during 2002-08, and that the grounding increased between 11/2004 and 12/2006. They propose that the MIT would have calved because of increasing grounding extent even if the tongue had not been hit by an iceberg. The authors also examine the rate of change in the area of MIT, and estimate an interval between subsequent tongue calving events of 70 years.

A quick summary of this review is that this manuscript needs a great deal of editing and extensive revision before it is ready for publication. I have presented some of my thoughts about what needs to be fixed, but the writing is of uneven quality and my comments about the reliability of interpolated data should (in my opinion) lead to substantial changes in the text and figures. With this in mind, I have not gone to the effort of editing the paper in detail, and hope that the authors can do so themselves.

**Reply: Thank you for your comments. We have made many revisions to the structure. Please find related changes highlighted in red in the text.**

A major problem in this manuscript is the lack of bathymetry data under the MIT. Data are scattered, at varying density, seaward of the ice front, and the bathymetry maps appear to resolve the seaward extent of the Mertz Bank, but under the tongue the maps are entirely based on interpolated values. This makes the maps in figure 5 and the statistics in table 2 hard to believe except at the very edge of MIT where the altimetry and bathymetry more or less coincide. The conclusions of the paper are largely independent of the data everywhere except in the area where the data are credible, which makes me wonder why the authors chose to show the mapped elevation-difference values in the areas for which they have no data. The authors should make a clear distinction between results derived from measurements and results derived from interpolated values, and the relative expected accuracies of each.

**Reply: Through further reading the references about GEBCO and ETOPO1 seafloor DEM, we do find there exists a bathymetry data gap under the MIT as the reviewer has pointed out. According to Beaman et al. (2011), the oldest bathymetry data collected along the margin of the MIT was at least from 2000. Thus, the boundary of the MIT in 2000 is used to identify bathymetry measurement gaps, as is indicated in Fig. 6. We want to use this boundary to identify the different quality of the seafloor DEM data.**

**As far as we know, there are no other new bathymetry measurements which can be used to verify the region of data gaps. Thus a further validation of the seafloor DEM is not conducted. Luckily, the ice tongue moved further into the ocean from 2000 to 2010 before calving, flowing over seafloor where bathymetry measurements density is good. Furthermore, the grounding detected is all located beyond the 2000 MIT boundary. Thus the analysis of grounding detection near the ice front in 2002, 2004, 2006 and 2008 is convincing. In this revision, the grounding detection result from Fig. 6 (Fig. 5, last version) is unchanged. However, the boundary of MIT in 2000 is now added to separate the seafloor area under the MIT where the seafloor was interpolated from the surrounding area where bathymetry measurements have been made. The statistical result of grounding detection in Table 2 is recalculated accordingly. Detailed discussion on the seafloor DEM is added in a newly added section 6.2.3.**

A second problem is that the methods are difficult to interpret, in large part because the report cited as "Wang 2014" is not readily available online, and the paper "Wang et al 2014" appears to describe a method for estimating freeboard change, rather than the absolute freeboard used in the present study. A good deal of the material in this paper is based on a technique in that report, described briefly in section 3.1(126-135). This paper should include a full description of the technique. In particular, it is not clear from the description how the relocation step works or what it is supposed to accomplish, or how the surface slope relates to errors in this relocation (line 241). I would also have liked to see a justification for the kriging interpolation between ICESAT profiles; the grounding features appear to be small compared to the gaps between ICESAT tracks, which makes me suspect that the krigged freeboard values may not provide a good indication of grounding.

**Reply: For freeboard production, we did not cite Wang 2014, but Wang et al. 2014, which did show how to use ICESat/GLAS data to produce a freeboard map in 2009 before MIT calving. In the revision, more details on the freeboard production method are given in Section 3.1. Uncertainty of kriging interpolation using ICESat/GLAS data is investigated as well which is about ±1.8 m on average in a new Section 4.1. The uncertainty of kriging interpolation is also considered when calculating the final grounding detection accuracy in Section 4.**

The authors should also be clear about the tidal values used in the freeboard study. Are the altimetry values corrected for tides? What is the "lowest sea level" mentioned at 155, and elsewhere? Is it derived from a tide model, or is it the lowest observed sea level? Is the tide model on the ICESAT product used, or is a different tide model used? What are the errors involved in each part of this?

**Reply: More details on ICESat/GLAS preprocessing and the method to produce the freeboard map has been given in the revised text. Tide correction of ICESat/GLAS GLA12 from TPX07.1 is removed to obtain the instantaneous sea surface condition. "lowest sea level" in line 155 may be confusing and has been changed to "lowest sea surface height among extracted sea surface height from different tracks and different campaigns, which is -3.35 m". It is derived by comparing all sea surface height derived from different tracks and campaigns from 2003 to 2009, not from a tide model. The lowest sea surface height stands for the lowest sea level around Mertz from 2003 to 2009 and is directly from ICESat/GLAS observation. Sea level lower than -3.35 m may in fact exist over the Mertz region since limited ICESat observation in any year may not catch the lowest one. The influence of sea level -3.35 m used in this study is discussed more in a new Section 6.2.1.**

The English in the manuscript needs improvement. A few idioms are used throughout that are confusing or distracting. "Inversed" should be "inverted." "Area-changing rate" should just be "area rate." "Least-square" should be "least-squares." Activities in the current study should be in present tense, citations to the literature should be in past tense.

**Reply: These grammatical issues have been resolved. The error or uncorrected expression pointed to by the reviewer has been corrected. More unclear descriptions or grammar misuses have been corrected as indicated in red in the revised text.**

The FAC calculation (3.2) has some nice features, but needs to be described in more detail. How is the least-squares inversion carried out? What are the error sources?

**Reply: More details about FAC calculation have been given. Please read Section 3.2 and 6.2.2.**

160-177- is the extensive discussion of other methods of calculating the FAC germain to this study? This section would be clearer if much of this were omitted.

**Reply: The introductory part of Section 3.2 has been revised. One paragraph not related to the FAC calculation much has been removed. Please read the revised Section 3.2.**

222-229- this paragraph should be in the introductory part of section 3.2, not after the calculations have been presented.

**Reply: In the introductory part of Section 3.2, the principle of FAC has been given. However, we want to use this text to discuss limitations of the FAC as calculated from these selected icebergs. In the revision, this paragraph in question is now moved to Section 6.2.2 as part of a deeper discussion on FAC extraction.**

247: Where are the interpolation errors for freeboard and bathymetry?

**Reply: The influence of kriging interpolation is discussed in Section 4.1. Also the uncertainty of kriging interpolation is derived and now considered in the final accuracy of grounding detection.**

**From Beaman et al. (2011), the poorest accuracy of single beam and multi-beam measurements was provided. Thus, we use it directly to evaluate the accuracy of the data in Section 4.1. Because the original bathymetry data product from Beaman et al. (2011) and Fretwell et al. (2013) was not collected and processed by us, it is impossible to evaluate the uncertainty of the products. As far as we know, there is no other new bathymetry measurements that can be used to evaluate the seafloor DEM. Thus, for seafloor DEM, we just use the poorest accuracy to reflect the uncertainty of seafloor DEM. The seafloor DEM is further discussed in a new Section 6.2.3.**

254- 50 times the average slope is still a very small number (0.6 degrees). A better estimate of the error due to crevassing would be to directly incorporate the crevasse depth into the calculation- thus, instead of v*slope_error (12 m) the contribution would be closer to 50 m.

**Reply: Crevasses are important features on the surface of the MIT. In the middle of the tongue, large crevasses can reach a depth of about 50 m. However, this is an extreme and rare occurrence. Around the ice front, the freeboard is about 30 m, as can be seen from Wang et al. (2014). It is therefore not proper to set the crevasse depth in that area to be as large as 50 m. The freeboard error caused by our approach is reasonable because we want to explore the average contribution to grounding detection from footprint relocation by considering ice velocity uncertainty and average surface slope. In this study, we have already magnified it by 50 times. As we feel this is a reasonable approach for the ice front treatment, we kept our original approach in the revised manuscript.**

256-62: Why do we need to consider the freeboard stable (or not stable?) It appears that only static estimates of freeboard are used here (derived from single ICESAT campaigns)- so why does it matter that there would be a change (or not) in the freeboard?

**Reply: Greater details about the method for freeboard extraction, relocation and mapping are added in Section 3.1. Because we use all ICESat/GLAS data from 2003 to 2009 to produce freeboard for different years, freeboard changes do matter if freeboard changed greatly. Thus,**

**the uncertainty of freeboard change rate does contribute to the final accuracy of grounding detection.**

257 "annual changing rate of freeboard" should be "annual rate of freeboard change" or "freeboard rate"

**Reply: Done**

273: What is the significance of Edif < 34 m? Based on 263-268, this would indicate "not extremely confidently identified as ungrounded." Wouldn't a better statistic be Edif < -34, or "Extremely confidently identified as ungrounded?"

**Reply: After considering the interpolation error, the accuracy for grounding detection is now ± 23 m (± 17 m in last version). We provide statistics for those elevation difference with E_dif less than 46 m (twice the standard deviation) so one can have a better estimate of grounding at the tongue. When using E_dif less than -46 m, the slightly grounded sections will be neglected. We use this value of 46 m to describe all possible grounding regions. Furthermore, the statistics in Table 2 do show results in different intervals from 46 m to less than 0.**

280: Again: Do you mean "less than -17 m?"

**Reply: We mean "less than 23 m" because in 2002, the minimum of 'E_dif' is larger than "-23 m". In this revision, '17' is changed to '23' because of a revised consideration of the kriging interpolation error.**

291-293: Reporting Edif within the tongue is a problem, since the bathymetry is not known there. You might report changes along the margin, but the statistics reported here don't seem to mean anything.

**Reply: We actually want to express the '$E_{dif}$' for those regions listed in Table 2 only, not all regions under the tongue. These regions with '$E_{dif}$' less than 46 m do fall beneath the ice front of the MIT. In this revision, we change it to "From 2002 to 2008, more regions under the MIT have '$E_{dif}$' less than 46 m the area of which increased from 8 km$^2$ to 17 km$^2$. Additionally, the mean of '$E_{dif}$' under the tongue for those having '$E_{dif}$' less than 46 m gradually decreases from 28.8 m to 12.3 m, according to which we can conclude that the ice front was grounded more significantly with passing time. "**

302 Combine the first two sentences, which form a joint conclusion: ". However," should be ", and that"

**Reply: Done.**

325 (and elsewhere) "Area-expanding trend" should be "area rate" or "rate of area change"

**Reply: Done.**

367-78: The significance of this paragraph is not clear. Ice-berg scouring is not discussed elsewhere in the paper, so the scientific question addressed by this paragraph needs more introduction.

**Reply: This section is removed.**

577- "is used in figure 6"– this appears not to be true.

**Reply: Changed it to 'Fig. 4'**

589: "closed' – should this be "closest?"

**Reply: Done.**

594: The legend here is not consistent with the caption.

**Reply: Legend is changed.**

606: It is hard to distinguish the outline from the "grounding part." The choice of colors (yellow on yellow) is not good.

**Reply: This figure is redrawn and yellow lines are not used in this version.**

Reviewer 2:

The authors use bathymetric, ICESat, and Landsat data products to estimate the firn air content, depth below sea level, re-grounding locations, and advance rate for the Mertz Ice Tongue from 2002-2008. They find that grounding along the Mertz Bank resulted in slight rotation and rifting of the Mertz Ice Tongue that would have resulted in the ice tongue's eventual collapse in the absence of any additional triggering mechanisms. Further, they suggest that the ice tongue collapse has a periodicity of ~70 years and that this periodicity results in periodic variations in local sea ice formation and bottom water formation. Although the topic of the manuscript is interesting, the limited presentation of the methods and irregular quality of the writing make it difficult to follow. In addition to the major revisions listed below, I recommend that the authors go through the text in detail to check the writing and to make sure that all figures are legible.

**Reply: Thanks for your comments. We have thoroughly revised the manuscript. The changes are highlighted in red in the revised text.**

1) In the data and methods sections, the authors frequently refer the readers to other publications rather than describe the data processing procedures in detail in the text. I find this to be particularly concerning for the freeboard inversions to estimate ice thickness because small errors in freeboard can lead to large variations in the estimated tongue depths. In order to have confidence in the provided tongue depths, I recommend including more detail on the relocation and interpolation procedures. Similarly, more information regarding the uncertainty of the bathymetry data used to identify grounded regions would be incredibly helpful.

**Reply: More details about freeboard map production using all available ICESat/GLAS data from 2003 to 2009 is added in a revised Section 3.1.  More discussion is added as well in a new Section 6.2.3.**

**From Beaman et al. (2011), the poorest accuracy of single beam and multi-beam measurements was provided.  Thus, we use it directly to evaluate the accuracy of this data in Section 4.  Because the original bathymetry data product from Beaman et al. (2011) and Fretwell et al. (2013) was not collected and processed by us, it is impossible to fully evaluate the uncertainty of the products.  The seafloor DEM is further discussed in the new Section 6.2.3.**

**We do acknowledge that in regions with bathymetry gaps, the quality of seafloor topography is poorer compared to other regions. According to Beaman et al. (2011), the oldest bathymetry data used to produce the seafloor DEM that are already known was at least from 2000. Thus, the boundary of the MIT in 2000 is used to identify bathymetry measurement gaps, as is indicated in Fig. 6. We use this boundary to identify the different quality of the seafloor DEM data since as far as we know, there is no other new bathymetry measurements**

that can be achieved to verify the region of data gaps. Luckily, the ice tongue moved further into the ocean from 2000 to 2010, before calving, into regions where bathymetry measurements are good. Furthermore, the grounding we detect is all located beyond the 2000 MIT boundary. Thus the analysis of grounding detection near ice front in 2002, 2004, 2006 and 2008 is convincing.

2) It's really difficult to follow the firn air content approximation. I assume the bed elevations are really well constrained under the targeted icebergs and you are simply iteratively estimating the iceberg depths for gradually decreasing values of the mean iceberg density. The units obtained for the firn air content estimated using this method require explanation. I assume that they represent the difference in iceberg depth assuming a constant ice density and the final ice density estimated through the comparison with the underlying bathymetry since the units are in meters, but this is not presented anywhere. It would be helpful to also present the final density inferred for the firn column so that it is easier to compare your estimates with other observations. The error estimates obtained for firn air content should also be presented in more detail. I am particularly concerned with the assumption that the ICESat tracks capture the thickest portion of each iceberg. I'd be more confident in the firn air content estimates if I was also shown that there are relatively small variations in iceberg freeboard along the ICESat tracks because that would increase confidence that the iceberg grounding location is captured by the ICESat data.

**Reply: The method for FAC calculation has been revised in Section 3.2. The seafloor DEM is well controlled by the bathymetry measurements as can be seen from S-Fig. 1. A paragraph on why FAC is used and how it is obtained is provided in Section 3.2. Some text not related so much to FAC calculation has now been removed. Fig. 9 is added to show the spatial distribution of freeboard of icebergs. More details on freeboard measurements from ICESat/GLAS, and the limitation of our method for FAC calculation is discussed in Section 6.2.2. Our estimated FAC around Mertz is compared with published modeling results (from Ligtenberg, 2014) in Section 6.2.2. Calculating the average density directly is beyond the scope of the current manuscript.**

3) The addition of the iceberg scour section at the end of the discussion is somewhat out of place with the rest of the manuscript. I suggest removing it entirely.

**Reply: This section is removed.**

---

## Referee Report (RR1)

**Review of "Grounding and Calving Cycle of Mertz Ice Tongue Revealed by Shallow Mertz Bank" by Wang et al. in The Cryosphere**

Line-by-line comments:
line 29: Replace "The calving of MIT can be cyclical because…" with "In the calving induced by iceberg collisions, our observations suggest that calving of the MIT is a cyclical process controlled by the presence…"
line 40: Change "… Mertz polynya, and sea-ice production and dense, shelf-water formation" to "…Mertz polynya, sea ice production, and dense shelf water formation…"
line 47: Change "how severe the grounding was" to something like "the extent of grounding" or possibly even "the severity of grounding"
line 60-61: This is a somewhat odd sentence "Grounding as a potential factor can affect the stability of an ice tongue…". I would suggest that you revise and rephrase to indicate how grounding can affect ice tongue stability, not just state that it can influence stability.
line 77: "launched" not "lunched"
line 130: Change to "The first step involves data processing…"
lines 134-135: "… are corrected following the procedures in Wang et al. (2012, 2013)."
line 137: If I'm interpreting this correctly, change to "… to obtain estimates of the instantaneous sea surface height." You talk about extracting sea surface elevations in the next paragraph, however, so I'm not sure that's what you mean here.
line 139: How are the data prepared for use?
line 150: What do you mean by "almost repeatedly"?
line 156: "For example, consider ICESat data from…"
line 158: "… distance between track T165 and T31 is ~7.5 km without accounting for ice advection between observation dates."
line 170: Remove "a little"
line 184: Replace "because of known ice tongue outlines from Landsat images" to "when the ice tongue outline can be delineated from Landsat images"
line 186: "… assuming hydrostatic equilibrium and using the lowest sea-surface height (further discussed in section 6.2.2), which is extracted from ICESat/GLAS data…"
line 201: Remove "at"
line 209: You don't need to define Edif here since you just defined it before the equation but you need to define Esf (which I assume is the sea floor elevation).
line 213-214: The author's name is "van den Broeke"
line 228: "that will influence the density of the ice tongue" rather than "that makes ice mass calculation complicated"
line 311: "compared with interpolated freeboard estimates"
line 325: Add a reference for the inter-campaign uncertainty in ICESat data
line 372: I recommend changing "was grounded more significantly" to "became more firmly grounded" because you aren't performing any statistical tests.
lines 399-400: Remove one of the "over this period"
line 411: End line 410 with a period after the reference then start a new sentence explaining that, based on your observations, you think only one large calving event occurred. ("Based on the interactions between the Mertz ice tongue and Mertz Bank

suggested by our observations and described below, it is likely that only one large calving event occurred between 1912-29156.")

line 425: "Because of the continuous advection of ice from upstream and the fixed location of the shallow Mertz Bank, the calving is…"

line 434: Of "the" Mertz polynya.

line 439-441: Why would a shorter ice tongue lead to a reduction in katabatic winds (which are driven by air temperature and pressure gradients over steep grounded ice) and polynya size? Elaborate.

line 443: "Mertz wellwhich"?

lines 512-514: I'm not sure I follow this. Why does the absence of shadowing (of ice, the ocean, ???) indicate flotation?

Figure 3: Change the color and/or relative position of the dashed-dotted line so that it is easier to discern.

---

## Author Response (AR2)

**We are grateful to the editor and reviewers for their time and constructive comments to improve this manuscript. Here we address our reply point by point, in bold font. Both editor's and reviewer's comments are in regular font. All changes are marked with red in the revised manuscript.**

Editor Decision: Reconsider after major revisions (15 Apr 2016) by Andreas Vieli

Comments to the Author:

Dear X. Wang et al,

Thank you for your substantial revision of your paper and for addressing the comments/concerns of the two reviewers. As the revisions were rather substantial i will send the paper out for review again. However, although the authors have also addressed most of editing and writing (English) issues highlighted by the authors, the revised parts and new text passages clearly suffer again from such editing/writing issues (singular/plural, missing articles, awkward/confusing formulations etc...). I have listed some below (the most obvious) but this list is not exhaustive. So, before I can send the paper out to review again, i ask you to carefully check the whole text again for editing issues and writing etc as we should not expect that the reviewers should do this editing job. I am sure the reviewers will very much appreciate your effort on this. Not that the list below is probably far from complete and there may be more issues that I did not spot when reading through quickly. So although the system may say 'major revision before reconsideration' it actually is rather 'minor' revisions (as easy to do) before sending it to re-review.

Thank you and best regards

Andreas Vieli

The editor

**Reply: Thank you for your comments. We have revised the manuscript thoroughly according to your comments.**

Some obvious editing/writing issues:

Line 129/130: '…for each of the datasets from Novemeber…'

**Reply: Done.**

Line 133: should be in plural: 'clouds'

**Reply: Done.**

Line 137: add 'the': '…correction from the TPX07.1…'

**Reply: Done.**

Line 139: 'under WGS-84 ellipsoid' seems awkward, maybe '…using the WGS-84 ellipsoid …' or '…with the WGS-84 ellipsoid …' would be better

**Reply:  We change it to 'related to the WGS-84 ellipsoid'.**

Line 148: i think the word 'even' is here rather awkward and is not needed here.

**Reply:  'even' is deleted.**

Line 151: I can not follow '…MIT almost repeatedly along the same track…'. Rephrase clearer.

**Reply: We change it to 'ICESat observed the MIT almost repeatedly along different tracks in different campaigns'.**

Line 173 and elsewhere: please remove the inverted commas around all the variables in the text (e.g. 'x'), writing them as italics is enough.

**Reply:  All the inverted commas around the variable in italics in the text have been removed.**

Line 173 to 178: whenever it say something like…'var1' and 'var2' is… I think you should use the plural for the verb: …'var1' and 'var2' are… and then the noun later should also be in plural (for example: t1 and t2 are the start and end times.

**Reply: All these issues have been corrected in the revision.**

Line 183: replace 'kriging' by 'it'

**Reply: Done.**

Line 203: replace '…will be soon introduced…' by '…is introduced…'

**Reply: Done.**

Line 218: change to '…convenient (…) and is defined…'

**Reply: Done.**

Line 226: delete the second 'layer' ('…of the firn layer on top of an …')

**Reply: Done.**

Line 234: simplify to 'It is critical to target and use icebergs that fulfil these requirements….'

**Reply: Done.**

Line 249-250: remove: in terms of estimating ….itself.

**Reply: Done.**

Line 295-296: move the 'and' to after 'Delta_E_krig'

**Reply: Done.**

Line 308: 'is relatively correct'??? I do not understand this, do you mean 'reasonably accurate'?

**Reply: We change it to 'reasonably accurate'.**

Line 320/321: 'the interpolated freeboard' rather than 'the freeboard interpolation'? AND 'in average'

**Reply: Done.**

Line 349/350: '…and outlines of the MIT … are shown…'

**Reply: Done.**

Line 364: add a comma after 'Also'. This sentence is as a whole a bit awkward.

**Reply: We change it to 'Also, it would be difficult for the MIT to approach the buffer region (indicated with yellow to red color in Fig. 6) as the surrounding Mertz Bank gets shallower and steeper, suggesting substantive grounding potentials'.**

Line 374: 'as time passed on' (rather than 'with passing time')

**Reply: Done.**

Line 390: awkward formulation: maybe write '…needed to climb more 140m to pass it…' or ' …needed to climb the 140m obstacle to cross it'.

**Reply: We change it to 'the MIT would have needed to climb the 140m obstacle to cross it'.**

Line 408: I think this should be Fig. 8 (not Fig 7)

**Reply: Done.**

Line 442: the '70 years' are in a random place in this sentence, maybe write: 'Additionally, the 70 year cycles of MIT…. well which make …'

**Reply: We change it to 'Additionally, the 70 year cycles of MIT calving coincides with surface ocean condition around Mertz well which makes the explanation much more compelling'.**

Line 450: '…are discussed in more depth/detail'

**Reply: Done.**

Line 456: where does the number -3.35 m come from?

**Reply: '-3.35 m' is from the Line 190, Section 3.1. It is derived from all sea level extractions from different tracks in different campaigns.**

Line 498-499: awkward formulation, maybe say: 'However, other time dependent modelling results from the Mertz region were close to …..' and delete: 'Our result is smaller'.

**Reply: Done.**

Line 475: somewhat awkward formulation: 'worked only several times a year..' change.

**Reply: We change it to 'Second, because ICESat/GLAS observed only several times a year on repeat tracks and icebergs was rotating slowly, the elevation profile in 2006 and 2008 along the same track T1289 may not come from the same ground surface. '.**

Line 476: add a 'the' before 'elevation'.

**Reply: Done.**

Line 481-482: awkward: maybe say: 'Since the larger freeboard measured in 2006 indicates ….'

**Reply: Done.**

Line 484: '…reason to select the measurement…'

**Reply: Done.**

Line 487: '…is critical to the final success…'

**Reply: Done.**

Line 489: '…along the margin…'

**Reply: Done.**

Line 492: '…front, for both the east and west flanks, …

**Reply: Done.**

Line 495: I do not see a white polygon in fig 6, do you mean the light grey dash-dotted line with label 20000125? Clarify and maybe change labels in fig 6 to the years (e.g. 2000).

**Reply: We change it to 'Inside of the MIT boundary of 2000, the closer to the dash-dotted polygon (Figs. 6 and 7), the better the accuracy the seafloor DEM.' Figs. 3, 6 and 7 are redrawn in order to make the 2000 boundary consistent with each other.**

Line 499-450: awkward formulation: '…but only the poorest accuracy …', rephrase

**Reply: We change it to 'However, from Beaman et al. (2011), no uncertainty on the seafloor DEM was systematically provided. Instead, only the poorest accuracy of single or multi-beam bathymetric measurements was available'.**

Line 500-502: awkward sentence, rephrase ('…it is not possible to conduct further work on evaluation of the seafloor bathymetry ……is difficult to assess'.

**Reply: Done.**

Line 504-507: I do not understand these sentences.

**Reply: We change it to 'Since Beaman et al. (2011) provided the most accurate seafloor DEM over Mertz according to our best knowledge, seafloor DEM inside of dash-dotted polygon (Fig. 7) is kept and the grounding detection is conducted there (Fig. 6) as well. Additionally, the ice tongue never stopped flowing further into the ocean, where the bathymetry measurements density is good'.**

Line 516: 'gradual' (not gradually)

**Reply: Done.**

Line 544: '…is a/the dominant factor….'

**Reply: We change it to 'Also the calving cycle of the MIT explains the observed cycle of sea surface conditions change well, which indicates the calving of the MIT is the dominant factor for sea sea-surface condition change'.**

Figure 5: caption of figure: please carefully check and rewrite caption, for example (a) does not show 'freeboard' but profile locations …

**Reply: We change it to 'Figure 5. Evaluation of kriging interpolation method over the MIT using freeboard data derived from ICESat/GLAS. (a) shows profile locations of freeboard derived from ICESat/GLAS after relocation over the MIT. Gray dots indicate ICESat/GLAS used for interpolation using kriging method. The blue dashed square indicates the region used to**

investigate interpolation accuracy of kriging method, about 7 km×7 km.  Inside of the square, freeboard data marked with green dots are used to check the accuracy of freeboard interpolated with kriging. (b) is the freeboard comparison result derived by subtracting krigged freeboard from freeboard derived from ICESat/GLAS. The spatial distribution and the histogram of freeboard difference are shown in the lower left and upper right respectively. The black polygon filled with light blue shows the boundary of MIT on November 14, 2002.'

Caption Fig 7Line 774: 'The grounding sections of the MIT boundary in …'

**Reply: Done.**

Supplementary figure 1: please check and rewrite caption of this figure very carefully:

Line 3: 'in THE red rectangle…'?

**Reply: We change it to 'The embedded figure in the lower left is the zoom in of the red rectangle which shows the positions of iceberg A and B (polygon filled in red) on February 19, 2008 (Fig. 4)'.**

Line 4: use date rather than 50th day

**Reply: We use 'on February 19, 2008' in the revision.**

Line 5: 'Blue polylines show the seafloor contours…'

**Reply: Done.**

Line 6 onwards: does this all have to go into the bracket? These are several sentences!!!

**Reply: The bracket has been removed.**

Line 7: 'However, for being able to use the Figures…..'???

**Reply: Done.**

Line 8: 'geo-registered'

**Reply: Done.**

Line 10: '…with THE thick gray line…'

**Reply: Done.**

Line 11: successful (not successfully)

**Reply: Done.**

Reviewer 1:

This paper uses altimetry and bathymetry data to map changes in the grounded area of the Mertz ice tongue during the early-to-mid 2000s. They use the surface heights of lightly-grounded icebergs to estimate the firn air content for the ice tongue, and use this, geoid-corrected altimetry measurements, and a bathymetry map of the ice shelf, to map the difference between the bottom of the ice shelf, at hydrostatic equilibrium, and the sea bed for different time periods. Areas where these maps show the (hydrostatic) ice bottom below the seabed are treated as grounded. The authors find that the northwest flank of MIT (Mertz Ice Tongue) was grounded during 2002-08, and that the grounding increased between 11/2004 and 12/2006. They propose that the MIT would have calved because of increasing grounding extent even if the tongue had not been hit by an iceberg. The authors also examine the rate of change in the area of MIT, and estimate an interval between subsequent tongue calving events of 70 years.

A quick summary of this review is that this manuscript needs a great deal of editing and extensive revision before it is ready for publication. I have presented some of my thoughts about what needs to be fixed, but the writing is of uneven quality and my comments about the reliability of interpolated data should (in my opinion) lead to substantial changes in the text and figures. With this in mind, I have not gone to the effort of editing the paper in detail, and hope that the authors can do so themselves.

**Reply: Thank you for your comments. We have made many revisions to the structure. Please find related changes highlighted in red in the text.**

A major problem in this manuscript is the lack of bathymetry data under the MIT. Data are scattered, at varying density, seaward of the ice front, and the bathymetry maps appear to resolve the seaward extent of the Mertz Bank, but under the tongue the maps are entirely based on interpolated values. This makes the maps in figure 5 and the statistics in table 2 hard to believe except at the very edge of MIT where the altimetry and bathymetry more or less coincide. The conclusions of the paper are largely independent of the data everywhere except in the area where the data are credible, which makes me wonder why the authors chose to show the mapped elevation-difference values in the areas for which they have no data. The authors should make a clear distinction between results derived from measurements and results derived from interpolated values, and the relative expected accuracies of each.

**Reply: Through further reading the references about GEBCO and ETOPO1 seafloor DEM, we do find there exists a bathymetry data gap under the MIT as the reviewer has pointed out. According to Beaman et al. (2011), the oldest bathymetry data collected along the margin of the MIT was at least from 2000. Thus, the boundary of the MIT in 2000 is used to identify**

**bathymetry measurement gaps, as is indicated in Fig. 6. We want to use this boundary to identify the different quality of the seafloor DEM data.**

**As far as we know, there are no other new bathymetry measurements which can be used to verify the region of data gaps. Thus a further validation of the seafloor DEM is not conducted. Luckily, the ice tongue moved further into the ocean from 2000 to 2010 before calving, flowing over seafloor where bathymetry measurements density is good.  Furthermore, the grounding detected is all located beyond the 2000 MIT boundary.  Thus the analysis of grounding detection near the ice front in 2002, 2004, 2006 and 2008 is convincing. In this revision, the grounding detection result from Fig. 6 (Fig. 5, last version) is unchanged. However, the boundary of MIT in 2000 is now added to separate the seafloor area under the MIT where the seafloor was interpolated from the surrounding area where bathymetry measurements have been made. The statistical result of grounding detection in Table 2 is recalculated accordingly.  Detailed discussion on the seafloor DEM is added in a newly added section 6.2.3.**

A second problem is that the methods are difficult to interpret, in large part because the report cited as "Wang 2014" is not readily available online, and the paper "Wang et al 2014" appears to describe a method for estimating freeboard change, rather than the absolute freeboard used in the present study. A good deal of the material in this paper is based on a technique in that report, described briefly in section 3.1(126-135). This paper should include a full description of the technique. In particular, it is not clear from the description how the relocation step works or what it is supposed to accomplish, or how the surface slope relates to errors in this relocation (line 241). I would also have liked to see a justification for the kriging interpolation between ICESAT profiles; the grounding features appear to be small compared to the gaps between ICESAT tracks, which makes me suspect that the krigged freeboard values may not provide a good indication of grounding.

**Reply: For freeboard production, we did not cite Wang 2014, but Wang et al. 2014, which did show how to use ICESat/GLAS data to produce a freeboard map in 2009 before MIT calving. In the revision, more details on the freeboard production method are given in Section 3.1. Uncertainty of kriging interpolation using ICESat/GLAS data is investigated as well which is about ±1.8 m on average in a new Section 4.1.  The uncertainty of kriging interpolation is also considered when calculating the final grounding detection accuracy in Section 4.**

The authors should also be clear about the tidal values used in the freeboard study. Are the altimetry values corrected for tides? What is the "lowest sea level" mentioned at 155, and elsewhere? Is it derived from a tide model, or is it the lowest observed sea level? Is the tide model on the ICESAT product used, or is a different tide model used? What are the errors involved in each part of this?

**Reply: More details on ICESat/GLAS preprocessing and the method to produce the freeboard map has been given in the revised text. Tide correction of ICESat/GLAS GLA12 from TPX07.1 is removed to obtain the instantaneous sea surface condition. "lowest sea level" in line 155 may be confusing and has been changed to "lowest sea surface height among extracted sea surface height from different tracks and different campaigns, which is -3.35 m".  It is derived by comparing all sea surface height derived from different tracks and campaigns from 2003 to 2009, not from a tide model. The lowest sea surface height stands for the lowest sea level around Mertz from 2003 to 2009 and is directly from ICESat/GLAS observation. Sea level lower than -3.35 m may in fact exist over the Mertz region since limited ICESat observation in any year may not catch the lowest one. The influence of sea level -3.35 m used in this study is discussed more in a new Section 6.2.1.**

The English in the manuscript needs improvement. A few idioms are used throughout that are confusing or distracting. "Inversed" should be "inverted." "Area-changing rate" should just be "area rate." "Least-square" should be "least-squares." Activities in the current study should be in present tense, citations to the literature should be in past tense.

**Reply:  These grammatical issues have been resolved. The error or uncorrected expression pointed to by the reviewer has been corrected. More unclear descriptions or grammar misuses have been corrected as indicated in red in the revised text.**

The FAC calculation (3.2) has some nice features, but needs to be described in more detail. How is the least-squares inversion carried out? What are the error sources?

**Reply: More details about FAC calculation have been given. Please read Section 3.2 and 6.2.2.**

160-177- is the extensive discussion of other methods of calculating the FAC germain to this study? This section would be clearer if much of this were omitted.

**Reply: The introductory part of Section 3.2 has been revised. One paragraph not related to the FAC calculation much has been removed. Please read the revised Section 3.2.**

222-229- this paragraph should be in the introductory part of section 3.2, not after the calculations have been presented.

**Reply: In the introductory part of Section 3.2, the principle of FAC has been given. However, we want to use this text to discuss limitations of the FAC as calculated from these selected icebergs. In the revision, this paragraph in question is now moved to Section 6.2.2 as part of a deeper discussion on FAC extraction.**

247: Where are the interpolation errors for freeboard and bathymetry?

**Reply: The influence of kriging interpolation is discussed in Section 4.1. Also the uncertainty of kriging interpolation is derived and now considered in the final accuracy of grounding detection.**

**From Beaman et al. (2011), the poorest accuracy of single beam and multi-beam measurements was provided. Thus, we use it directly to evaluate the accuracy of the data in Section 4.1. Because the original bathymetry data product from Beaman et al. (2011) and Fretwell et al. (2013) was not collected and processed by us, it is impossible to evaluate the uncertainty of the products. As far as we know, there is no other new bathymetry measurements that can be used to evaluate the seafloor DEM. Thus, for seafloor DEM, we just use the poorest accuracy to reflect the uncertainty of seafloor DEM. The seafloor DEM is further discussed in a new Section 6.2.3.**

254- 50 times the average slope is still a very small number (0.6 degrees). A better estimate of the error due to crevassing would be to directly incorporate the crevasse depth into the calculation- thus, instead of v*slope_error (12 m) the contribution would be closer to 50 m.

**Reply: Crevasses are important features on the surface of the MIT. In the middle of the tongue, large crevasses can reach a depth of about 50 m. However, this is an extreme and rare occurrence. Around the ice front, the freeboard is about 30 m, as can be seen from Wang et al. (2014). It is therefore not proper to set the crevasse depth in that area to be as large as 50 m. The freeboard error caused by our approach is reasonable because we want to explore the average contribution to grounding detection from footprint relocation by considering ice velocity uncertainty and average surface slope. In this study, we have already magnified it by 50 times. As we feel this is a reasonable approach for the ice front treatment, we kept our original approach in the revised manuscript.**

256-62: Why do we need to consider the freeboard stable (or not stable?) It appears that only static estimates of freeboard are used here (derived from single ICESAT campaigns)- so why does it matter that there would be a change (or not) in the freeboard?

**Reply: Greater details about the method for freeboard extraction, relocation and mapping are added in Section 3.1. Because we use all ICESat/GLAS data from 2003 to 2009 to produce freeboard for different years, freeboard changes do matter if freeboard changed greatly. Thus, the uncertainty of freeboard change rate does contribute to the final accuracy of grounding detection.**

"annual changing rate of freeboard" should be "annual rate of freeboard change" or "freeboard rate"

**Reply: Done**

273: What is the significance of Edif < 34 m? Based on 263-268, this would indicate "not extremely confidently identified as ungrounded." Wouldn't a better statistic be Edif < -34, or "Extremely confidently identified as ungrounded?"

**Reply: After considering the interpolation error, the accuracy for grounding detection is now ± 23 m (± 17 m in last version). We provide statistics for those elevation difference with E_dif less than 46 m (twice the standard deviation) so one can have a better estimate of grounding at the tongue. When using E_dif less than -46 m, the slightly grounded sections will be neglected. We use this value of 46 m to describe all possible grounding regions. Furthermore, the statistics in Table 2 do show results in different intervals from 46 m to less than 0.**

280: Again: Do you mean "less than -17 m?"

**Reply: We mean "less than 23 m" because in 2002, the minimum of 'E_dif' is larger than "-23 m". In this revision, '17' is changed to '23' because of a revised consideration of the kriging interpolation error.**

291-293: Reporting Edif within the tongue is a problem, since the bathymetry is not known there. You might report changes along the margin, but the statistics reported here don't seem to mean anything.

**Reply: We actually want to express the '$E_{dif}$' for those regions listed in Table 2 only, not all regions under the tongue. These regions with '$E_{dif}$' less than 46 m do fall beneath the ice front of the MIT. In this revision, we change it to "From 2002 to 2008, more regions under the MIT have '$E_{dif}$' less than 46 m the area of which increased from 8 km$^2$ to 17 km$^2$. Additionally, the mean of '$E_{dif}$' under the tongue for those having '$E_{dif}$' less than 46 m gradually decreases from 28.8 m to 12.3 m, according to which we can conclude that the ice front was grounded more significantly with passing time. "**

Combine the first two sentences, which form a joint conclusion: ". However," should be ", and that"

**Reply: Done.**

(and elsewhere) "Area-expanding trend" should be "area rate" or "rate of area change"

**Reply: Done.**

367-78: The significance of this paragraph is not clear. Ice-berg scouring is not discussed elsewhere in the paper, so the scientific question addressed by this paragraph needs more introduction.

**Reply: This section is removed.**

577- "is used in figure 6"– this appears not to be true.

**Reply: Changed it to 'Fig. 4'**

589: "closed' – should this be "closest?"

**Reply: Done.**

594: The legend here is not consistent with the caption.

**Reply: Legend is changed.**

606: It is hard to distinguish the outline from the "grounding part." The choice of colors (yellow on yellow) is not good.

**Reply: This figure is redrawn and yellow lines are not used in this version.**

Reviewer 2:

The authors use bathymetric, ICESat, and Landsat data products to estimate the firn air content, depth below sea level, re-grounding locations, and advance rate for the Mertz Ice Tongue from 2002-2008. They find that grounding along the Mertz Bank resulted in slight rotation and rifting of the Mertz Ice Tongue that would have resulted in the ice tongue's eventual collapse in the absence of any additional triggering mechanisms. Further, they suggest that the ice tongue collapse has a periodicity of ~70 years and that this periodicity results in periodic variations in local sea ice formation and bottom water formation. Although the topic of the manuscript is interesting, the limited presentation of the methods and irregular quality of the writing make it difficult to follow. In addition to the major revisions listed below, I recommend that the authors go through the text in detail to check the writing and to make sure that all figures are legible.

**Reply: Thanks for your comments. We have thoroughly revised the manuscript. The changes are highlighted in red in the revised text.**

1) In the data and methods sections, the authors frequently refer the readers to other publications rather than describe the data processing procedures in detail in the text. I find this to be particularly concerning for the freeboard inversions to estimate ice thickness because small errors in freeboard can lead to large variations in the estimated tongue depths. In order to have confidence in the provided tongue depths, I recommend including more detail on the relocation and interpolation procedures. Similarly, more information regarding the uncertainty of the bathymetry data used to identify grounded regions would be incredibly helpful.

**Reply: More details about freeboard map production using all available ICESat/GLAS data from 2003 to 2009 is added in a revised Section 3.1. More discussion is added as well in a new Section 6.2.3.**

**From Beaman et al. (2011), the poorest accuracy of single beam and multi-beam measurements was provided. Thus, we use it directly to evaluate the accuracy of this data in Section 4. Because the original bathymetry data product from Beaman et al. (2011) and Fretwell et al. (2013) was not collected and processed by us, it is impossible to fully evaluate the uncertainty of the products. The seafloor DEM is further discussed in the new Section 6.2.3.**

**We do acknowledge that in regions with bathymetry gaps, the quality of seafloor topography is poorer compared to other regions. According to Beaman et al. (2011), the oldest bathymetry data used to produce the seafloor DEM that are already known was at least from 2000. Thus, the boundary of the MIT in 2000 is used to identify bathymetry measurement gaps, as is indicated in Fig. 6. We use this boundary to identify the different quality of the seafloor DEM data since as far as we know, there is no other new bathymetry measurements**

that can be achieved to verify the region of data gaps. Luckily, the ice tongue moved further into the ocean from 2000 to 2010, before calving, into regions where bathymetry measurements are good.  Furthermore, the grounding we detect is all located beyond the 2000 MIT boundary. Thus the analysis of grounding detection near ice front in 2002, 2004, 2006 and 2008 is convincing.

2) It's really difficult to follow the firn air content approximation. I assume the bed elevations are really well constrained under the targeted icebergs and you are simply iteratively estimating the iceberg depths for gradually decreasing values of the mean iceberg density. The units obtained for the firn air content estimated using this method require explanation. I assume that they represent the difference in iceberg depth assuming a constant ice density and the final ice density estimated through the comparison with the underlying bathymetry since the units are in meters, but this is not presented anywhere. It would be helpful to also present the final density inferred for the firn column so that it is easier to compare your estimates with other observations. The error estimates obtained for firn air content should also be presented in more detail. I am particularly concerned with the assumption that the ICESat tracks capture the thickest portion of each iceberg. I'd be more confident in the firn air content estimates if I was also shown that there are relatively small variations in iceberg freeboard along the ICESat tracks because that would increase confidence that the iceberg grounding location is captured by the ICESat data.

**Reply: The method for FAC calculation has been revised in Section 3.2. The seafloor DEM is well controlled by the bathymetry measurements as can be seen from S-Fig. 1. A paragraph on why FAC is used and how it is obtained is provided in Section 3.2.  Some text not related so much to FAC calculation has now been removed. Fig. 9 is added to show the spatial distribution of freeboard of icebergs. More details on freeboard measurements from ICESat/GLAS, and the limitation of our method for FAC calculation is discussed in Section 6.2.2. Our estimated FAC around Mertz is compared with published modeling results (from Ligtenberg, 2014) in Section 6.2.2.   Calculating the average density directly is beyond the scope of the current manuscript.**

3) The addition of the iceberg scour section at the end of the discussion is somewhat out of place with the rest of the manuscript. I suggest removing it entirely.

**Reply: This section is removed.**

[revised manuscript text omitted]

---

## Author Response (AR3)

We are grateful to the editor and reviewers for their time and constructive comments to improve this manuscript. Here we address our reply point by point, in bold font. Both editor's and reviewer's comments are in regular font. All changes are marked with red or green in the revised manuscript.

Editor Decision: Publish subject to minor revisions (Editor review) (18 May 2016) by Andreas Vieli

Comments to the Author:

Editor decision after reviews of revised revision.

Dear X. Wang et al,

The revised version of the manuscript was sent to 2 reviewers again (comments further below), and although they both indicated that the revised version has improved to some degree (compared to the original submission) they both stated that the general writing and English language clearly need further improvement, a point that I also made to you after the first revised version of the manuscript. I then already asked you to carefully edit and correct the whole document again, and not just the few points that I spotted. It is really not the job of the reviewers to correct the manuscript for English language grammatical errors, irregular punctuation, and awkward phrasing, but the author's. Both reviewers seemed rather disappointed regarding this aspect of the revised version.

Reviewer 2 had beside the English languages issues some rather minor further points to address (see list of reviewer 2), of which some concern the definitions/explanation (e.g. sea floor elevation, E_sf) in the methods.

Reviewer 1 was less positive and had besides the major language issues some rather substantial remaining issues concerning the methods and presentation (see major comments by reviewer 1) and on this basis recommended to reject this paper. As an editor I agree in principle with some of the more substantial points made by reviewer 1 but I think they are relatively easy to be addressed by the authors and do not fully justify a rejection.

 So, I suggest to the authors to very carefully address the

 (i) To very carefully address ALL the minor comments and points listed by the reviewers 1 and 2

**Reply: We have revised the manuscript thoroughly according to comments from both reviewers. All changes are marked with red or green in the revised manuscript.**

 (ii) Re-check and correct the ENTIRE publication again VERY CAREFULLY for English language, grammatical errors, awkward phrasing and small editing issues. I advise you to get the whole document checked again at the end by a native English speaker.

**Reply: We read and check the manuscript thoroughly again and the changes are marked with red or green in this revision.**

 (ii) To address the following more substantial points (a-d) by reviewer 1:

(a) Reviewer 1 is concerned about the calculation of E_dif below the substantive areas of MIT where no bathymetric data are available (only extra/interpolated). While such an E_diff can (theoretically) be calculated using the interpolated bathymetric data, I agree with the reviewer that it has not that much meaning other than based on the interpolated data (and other indicators) it is likely that MIT is floating there. There is a supplementary figure now showing where there is actually bathymetric data which helps regarding this aspect, but I would try to make this point a bit clearer in the main paper. There are a variety of ways to do this, here some suggestions:

-Provide the justification and all the available evidence for believing 'that the tongue is floating' rather than to carry out and show the entire calculation for E_dif based on extrapolated sea-floor heights.

-or clearly mark in fig 3 and in fig 6 (maybe with black hatching or similar) the areas where there is no bathymetric data available (or is just extrapolated).

-or show supplementary figure 1a) as Figure 3b).

Whatever option is taken, in the text it should be made clearer that the E_dif

**Reply: We agree with you on this point. We take up your third suggestion and the S-Fig 1a and S-Fig 1b from last revision is moved to the text as Fig 3b and Fig 3c. In this way, the spatial distribution of bathymetry is clearer.**

(b) regarding the detection of grounding from E_dif: I agree with reviewer 2 that the statement on line 340/341 of '...E_def less than 23 m corresponds to a very robust grounding event...' seems not quite right and also not consistent with the figures and interpretation in section 5 (line 353: E_dif less than 23m interpreted as ,almost grounded'). Maybe there is just a minus sign missing in front of 23m on line 340, I think it should say: below -23m it seems very strongly grounded, between -23m and 23m slightly grounded and above 23m unlikely to be grounded. Please check this carefully and adjust accordingly in whole document.

**Reply: We agree with you on this point. E_dif below -23 m should be strongly grounded. We have revised the manuscript thoroughly according to this comment.**

(c) Address the more structural issues (methodological explanations to the methods, discussion points to discussion (see detailed comments by reviewer 2).

**Reply: We have revised the manuscript thoroughly according to comments from both reviewers.**

(d) Clarify the point on applying the firn-air content calculation on heavily grounded ice bergs. I assume although you use 'heavily' grounded they are still very close to floatation but hardly move and hence flotation still is applicable.

**Reply: We didn't use "heavily grounded" icebergs for FAC calculation. Because the icebergs we chose could still move slowly, as can be seen from S-Fig. 1. We consider they are still close to floatation. The hydrostatic equilibrium still applies for these icebergs. "Heavily" was only**

**used to describe the B-9B iceberg which stayed in a point for several years. We have checked the manuscript thoroughly to make sure proper description on these icebergs.**

Thank you and best regards

Andreas Vieli

The editor

May 2016

Comments Reviewer 1:

This is my second review of this paper. I am somewhat disappointed with the revisions, in that one of the key points in my, and the other reviewer's, review was that the paper needed careful editing by the authors before it would be publishable. It appears that some of the edits suggested by the reviewers and the editor are included in this version, but that the authors have not found the grammatical errors, irregular punctuation, and awkward phrasing that were not specifically identified during the first review process. These remain in the present manuscript and it is the authors' responsibility to correct them.

**Reply: We have revised the manuscript thoroughly according to your comments. All changes are marked with red or green in the revised manuscript.**

The paper also remains somewhat disorganized. The discussion section contains material that belongs in the methods section, and the results section contains material that belongs in the discussion section. I have made some notes to this effect, but the authors should revise the paper carefully to make some that each section contains only the appropriate material.

**Reply: We have made proper changes on structure and revised the manuscript thoroughly according to your comments.**

For the scientific content of the paper, I am concerned to see that the authors are still presenting their calculation of E_dif under the MIT in areas where there are no measurements of bathymetry. There is a small area near the northwest edge of the tongue where the MIT has overrun some measured bathymetry, and in these areas it is appropriate to calculate E_dif, but for the rest of the tongue, it seems cleaner to provide a justification for believing that the tongue is floating than to carry out the calculation based on extrapolated sea-floor heights.

**Reply: In the last revision, we used the ice tongue boundary from 2000 to identify the data gaps. One can easily identify the data gaps using this boundary. To better illuminate the spatial distribution of bathymetry, we have moved the S-Fig. 1 in the last revision to text as Fig. 3b and Fig. 3c by taking up Editor's suggestion.**

The firn-air content calculation does not seem very useful. To calculate the firn air, the authors must use a point where the bathymetry is known, so that the ice bergs are grounding. But if the bergs are grounded, the hydrostatic approximation does not apply.

**Reply: The FAC calculation is useful to invert the ice draft and ice bottom elevation. We did select the icebergs located in region with bathymetry known, as can be seen from Fig. 3b. Although some icebergs were chosen as slightly grounding, they still moved slowly. From this point of view, hydrostatic approximation still applies.**

Last, the authors seem to have the detection limits for their grounding detection wrong. E_dif is equal to the inferred elevation of the ice bottom minus the sea floor. Suppose sea floor is known to be at -500 m, and the bottom of the ice berg is inferred to be at -524 m, give or take 23 m. Then E_diff=-24, with a one-sigma range between -47 and -1 meters. Likewise, if the bottom of the ice berg is inferred to be at -478 m, give or take 23 m, then E_diff =+22, with a 1-sigma range between -1 and 45 meters. The first case represents robust grounding, the second case In section 6, the authors say :

"E_diff less than 23 m corresponds to a very robust grounding event".

A robust grounding event should be one where the one-sigma range is entirely below zero ( my first case ) while a plausible, but not robust, grounding even is one where the one-sigma range includes zero, but also includes positive values (my second case). If E_diff is greater than 23 m, then at one sigma you can be confident that there is no grounding.

**Reply: We agree with you on this point. To make it further clearer, we add some sentences in the text. We take up Editor's suggestion and the standard to tell grounding or floating of an iceberg using E_diff is:  below -23m very strongly grounded, between -23m and 23m slightly grounded and above 23m unlikely to be grounded. We have revised the manuscript thoroughly according to this comment.**

87, Delete sentence beginning "With billions…" The other data types are not relevant here.

**Reply:We think that it is proper to keep it so readers can know there are different products from ICESat/GLAS and what data we are using for this study.**

132- This seems wrong. Clouds cannot cause ICESAT saturation.

**Reply: We change "the occurrence of clouds" to "high reflected natural surface".**

136: You subtract off TPX07.1, but do you reapply it in the freeboard calculation? The freeboard is the ice surface height minus the sea-surface height, and if you don't use a tide model, the tidal elevation causes an error.

**Reply: We had made it very clear in the last revision that we use instantaneous sea surface height to extract the freeboard. This is correct because freeboard is distance from ice top to sea surface, not sea level. The instantaneous sea surface height varies with tide, thus no need to consider the tide height.**

138-139: Not sure what this means. Please explain or delete.

**Reply: Elevation we used is usually referred to a geoid or ellipsoid. This sentence is to introduce the elevation we used in this study. The full name of WGS-84 and EGM08 can be found from line 105 to line 106. In this revision, we keep it unchanged in this revision and one reference is added on EGM08 in line 117.**

158-166: How are height gradients taken into account in this calculation? For a sloping surface, relocating the footprints by the ice velocity will introduce a spurious elevation change signal.

**Reply: Because of sloping surface of the MIT, sloping error caused by footprint relocation must be considered and put into the final error sources. In this study, the contribution of**

**footprint relocation to freeboard uncertainty is calculated as "Δd" multiplying "Δs", where "Δd" is the relocation error caused by the uncertainty of ice flow, "Δs" the average surface slope of the MIT.**

170: These equations need to be placed immediately after their introductory sentence. I suggest deleting the reference at 165 to these equations and adding one sentence such as "The corrected positions of the footprints are:" at 170.

**Reply: We take up your suggestion and move the equations 2 and 3 ahead so they connect to line 165 directly. We keep other description unchanged.**

179: don't need to discuss interpolation techniques not used here.

**Reply: We delete the sentence in this revision.**

186: Put equation 4 here, not eight lines below.

**Reply: We move equation 4 just after this sentence.**

253-56: How are the PIG icebergs relevant here? I suggest deleting the reference.

**Reply: We take up your suggestion and delete it in this revision.**

266-68: If the iceberg is grounded, then the bed is supporting some part of its weight. This means that it is not in hydrostatic equilibrium: rho_w D < rho_i (H_f +D –FAC). This seems to introduce a potentially large error into your calculation of FAC.

**Reply: We use "grounded" to describe the iceberg used for FAC calculation in the last revision because they moved slowly compared free drifting icebergs. However those icebergs could move as can be seen from S-Fig 1 which indicates that they are still very close to floatation and flotation still is applicable. We change "grounded" to "slightly grounded" in this revision when describing these icebergs used for FAC calculation.**

265-270: Are you using all the ice bergs, or only the 2006 measurements? If the latter, is this a least-squares calculation, or just a simple solution?

**Reply: We use the 2006 measurements to invers the FAC which was clearly addressed in line 263-266: "In this study, only the top two largest freeboard measurements of icebergs 'A' and 'C' from T1289 in 2006 are employed to calculate the FAC with Eq. (7) with a least-squares method under hydrostatic equilibrium". Because four equations were created, FAC is a least-squares solution.**

331: 50 times 0.00024 is 0.012. This is not a realistic estimate of the surface slope due to rugged, crevassed ice-tongue surfaces.

**Reply: The average slope of MIT was calculated as 0.00024 (Wang et al. 2014). In this study, we have already magnified it by 50 times. Because we want to explore the average contribution to grounding detection from footprint relocation by considering ice velocity**

uncertainty and average surface slope, not under an extreme situation, we feel this is a reasonable approach for the ice front treatment. The freeboard error caused by our approach is reasonable and we kept our original approach in the revised manuscript.

384-395: this material belongs in the 'Discussion' section.

**Reply: Done.**

Section 6.1:

Why does it make sense to talk about the area rate rather than the longitudinal flow rate? The mechanism proposed for interaction between MIT and the Mertz bank is that the MIS should break after it hits the bank. The time for the MIT to reach the bank after it calves is then distance between the end of the MIT and the bank divided by the speed of the end of the tongue. Casting this in terms of area seems to make the calculation more confusing, and is not clearly more accurate.

**Reply: We did not mean that MIT should break once it hits the Mertz Bank. Instead this is a slow progress, that's why we use the maximum ice tongue area and area rate to calculate the calving cycle. As can be seen from Massom et al. (2015), large rift could occur because of this hit and only the large rift propagates to the other flank can the ice tongue break off.**

Section 6.2: Most of this material belongs in the 'methods' section, where you should explain the limitations of, and the rationale for, your methods. It also sounds here like you are solving for the FAC using only the 2006 ice-berg heights, while before it sounded like you were including all the data in a least-squares calculation.

**Reply: Section 6.2.1 about the lowest sea surface height extraction is moved to method. Section 6.2.2 about FAC extraction is moved to section 3.2. For FAC calculation, we have made it very clear in line 263-266 in the last revision: "In this study, only the top two largest freeboard measurements of icebergs 'A' and 'C' from T1289 in 2006 are employed to calculate the FAC with Eq. (7) with a least-squares method under hydrostatic equilibrium". We keep this consistent throughout the manuscript.**

484: The discussion of the accuracy of the bathymetry belongs in the data section.

**Reply: The first paragraph about the accuracy of bathymetry is moved to the data section. However the other paragraphs about interpolation error are not proper to be moved to the data section, which are still kept in the discussion.**

497: Talking about the accuracy of the seafloor DEM under MIT makes little sense, because there are no measurements there, and the values provided are extrapolated/interpolated from measurement locations that are often far away. The arguments (503 - 517) that the bulk of MIS is floating are a better approach than making unfounded assumptions about the height of the bed where no data are available.

**Reply: This paragraph started from line 497 is removed in this revision.**

Figure 3: There are too many colors here. The bathymetry needs to be in a color scale that does not overlap the colors chosen for the outlines. Consider grayscale.

**Reply: Figure 3 is redrawn and grayscale is used for seafloor topography.**

Figure 6: Indicate clearly the areas in which the seafloor elevations are constrained by data. Do not show E_diff where they are not.

**Reply: We move S-Figure 1 to the text as Fig. 3b and Fig. 3c so that the spatial distribution of bathymetric measurements is clear.**

Figure 9: this figure is out of order, and is probably unnecessary.

**Reply: We want to use this figure to show freeboard of the icebergs and we added this figure in the last revision because reviewer 2 wanted us to show more about the iceberg freeboard. In this revision, we move this figure to supplementary.**

Comments Reviewer 2:

The incorporated revisions have considerably improved the manuscript. A few minor revisions are listed below.

line 29: Replace "The calving of MIT can be cyclical because…" with "In the calving induced by iceberg collisions, our observations suggest that calving of the MIT is a cyclical process controlled by the presence…"

**Reply: Done.**

line 40: Change "… Mertz polynya, and sea-ice production and dense, shelf-water formation" to "…Mertz polynya, sea ice production, and dense shelf water formation…"

**Reply: Done.**

line 47: Change "how severe the grounding was" to something like "the extent of grounding" or possibly even "the severity of grounding"

**Reply: Done.**

line 60-61: This is a somewhat odd sentence "Grounding as a potential factor can affect the stability of an ice tongue…". I would suggest that you revise and rephrase to indicate how grounding can affect ice tongue stability, not just state that it can influence stability.

**Reply: we change it to "Grounding as a potential factor can affect the stability of an ice tongue by possibly holding the tongue to delay calving (Massom et al. 2015)" in this revision.**

line 77: "launched" not "lunched"

**Reply: Done.**

line 130: Change to "The first step involves data processing…"

**Reply: Done.**

lines 134-135: "… are corrected following the procedures in Wang et al. (2012, 2013)."

**Reply: Done.**

line 137: If I'm interpreting this correctly, change to "… to obtain estimates of the instantaneous sea surface height." You talk about extracting sea surface elevations in the next paragraph, however, so I'm not sure that's what you mean here.

**Reply: we change it to "Furthermore, tidal correction from the TPX07.1 tide model in GLA12 data record is removed to obtain estimates of the instantaneous sea surface height". Also we change "sea level" to "sea surface height" thoroughly in this revision because we use the instantaneous measurements of sea surface height which is not "sea level".**

line 139: How are the data prepared for use?

**Reply: we change it to "Finally, elevation data related to the WGS-84 ellipsoid and EGM 08 geoid for ICESat/GLAS from 2003 to 2009 is ready for subsequent use."**

line 150: What do you mean by "almost repeatedly"?

**Reply: ICESat/GLAS did not have an exactly repeated ground track. For the same track, ground measurements can bias by several tens to several hundred meters in cross-track direction. That is why we use "almost repeatedly".**

line 156: "For example, consider ICESat data from…"

**Reply: Done.**

line 158: "… distance between track T165 and T31 is ~7.5 km without accounting for ice advection between observation dates."

**Reply: Done.**

line 170: Remove "a little"

**Reply: Done.**

line 184: Replace "because of known ice tongue outlines from Landsat images" to "when the ice tongue outline can be delineated from Landsat images"

**Reply: Done.**

line 186: "… assuming hydrostatic equilibrium and using the lowest sea-surface height (further discussed in section 6.2.2), which is extracted from ICESat/GLAS data…"

**Reply: Done.**

line 201: Remove "at"

**Reply: Done.**

line 209: You don't need to define Edif here since you just defined it before the equation but you need to define Esf (which I assume is the sea floor elevation).

**Reply: Done.**

line 213-214: The author's name is "van den Broeke"

**Reply: Done.**

line 228: "that will influence the density of the ice tongue" rather than "that makes ice mass calculation complicated"

**Reply: Done.**

line 311: "compared with interpolated freeboard estimates"

**Reply: Done.**

line 325: Add a reference for the inter-campaign uncertainty in ICESat data

**Reply: We change it to "we use ±0.15 m (Zwally et al. 2002) as the uncertainty of elevation data ($\varepsilon E_{sl}$)" in this revision.**

line 372: I recommend changing "was grounded more significantly" to "became more firmly grounded" because you aren't performing any statistical tests.

**Reply: Done.**

lines 399-400: Remove one of the "over this period"

**Reply: Done.**

line 411: End line 410 with a period after the reference then start a new sentence explaining that, based on your observations, you think only one large calving event occurred. ("Based on the interactions between the Mertz ice tongue and Mertz Bank suggested by our observations and described below, it is likely that only one large calving event occurred between 1912-29156.")

**Reply: Done.**

line 425: "Because of the continuous advection of ice from upstream and the fixed location of the shallow Mertz Bank, the calving is..."

**Reply: Done.**

line 434: Of "the" Mertz polynya.

**Reply: Done.**

line 439-441: Why would a shorter ice tongue lead to a reduction in katabatic winds (which are driven by air temperature and pressure gradients over steep grounded ice) and polynya size? Elaborate.

**Reply: We did not mean that a shorter ice tongue leads to a reduction of katabatic winds. However we mean that a shorter ice tongue leads to a small polynya size formed by katabatic wind. Different length of an ice tongue can block sea ice drifting from one side differently. A long MIT help to maintain a large polynya because more sea ice formed on the east side could not drift to the west side. With the effect of katabatic wind, sea ice produced from the west side is blown seaward. In this revision, we make this point much clear by explaining more.**

line 443: "Mertz wellwhich"?

**Reply: There should be a space between "well" and "which". We change it to "…Mertz well which…" in this revision.**

lines 512-514: I'm not sure I follow this. Why does the absence of shadowing (of ice, the ocean, ???) indicate flotation?

**Reply: To avoid confusion, we delete these sentences.**

Figure 3: Change the color and/or relative position of the dashed-dotted line so that it is easier to discern.

**Reply: Figure 3 is redrawn and grayscale is used for seafloor topography.**

[revised manuscript text omitted]

---

## Author Response (AR4)

**We are grateful to the Editor for his time and constructive comments to improve this manuscript. Here we address our reply point by point, in bold font. All the Editor's comments are in regular font. All changes are marked with red in the revised manuscript.**

Editor Decision: Publish subject to minor revisions (Editor review) (20 Jun 2016) by Andreas Vieli

Comments to the Author:

Dear authors,

The re-revised version addressed a lot of the issues listed by the reviewers including the more substantial ones of the missing bathymetric data under the main tongue, the structural issues and the corrections in the explanations for E_diff.

Unfortunately, there are still a lot of editing issues and awkward wording, a lot of them are in new or rewritten text parts and further the restructuring introduced some additional issues in explaining the methods. So despite having addressed most raised points by the reviewers the language, writing and wording remains unsatisfactory if not even sloppy, and gives the impression that the authors did not take the very clear instruction of the editor of 're-checking and correcting the ENTIRE publication for English language, grammatical errors, awkward phrasing and small editing issue' very seriously.

Again, it is not the editors or the reviewer's job to do this. I understand that English is not the first author's mother tongue but some of the co-authors or another native English speaker could maybe be asked for help.

I spent a lot of time going through this document now, and listed whatever I saw or struggled with in detail below, but I may likely not have spotted all and it is probably still not perfect. I give the authors here another chance to correct, tidy and check the whole document, but I asked them to very carefully and thoroughly do these revisions.

**Reply: We have read and checked the manuscript thoroughly to make the English language correct and the changes are marked with red in this revision.**

Specific comments/issues to be addressed:

To clarify where bathymetric data exists, in Fig. 3 the ship tracks with bathymetric data are now shown as (b) and (c) (from supplement), however, the scale is very different compared to (a).

For easing the readability of theses figures and clarification, I would suggest to show these ship tracks for exactly the same frame/area as in (a) (all the tracks outside are really not relevant).

**Reply: Figs 3b and 3c have been redrawn using the same frame and scale as what was used in Fig 3a.**

p. 2 Line 14: I would remove 'some' here.

**Reply: Done.**

p. 2, line 29/30: 'In the calving induced by iceberg collision' is an awkward formulation, the iceberg collision has nothing to do with the cyclic calving, so I would say: ' Our observation suggest that the calving of the MIT is a cyclical…'.

**Reply: Done.**

p. 2 line 32/33: Make clear that you refer to work from orther studies here, e.g. '…This calving cycle also explains the cyclic variations in sea-surface conditions around the Mertz detected by earlier studies.'

**Reply: Done.**

p. 3 line 55/56: awkward wording '…allows the causes ….gradually come into focus…' . I would rather say: '…allows to investigate the mechanisms of ice tongue instability and calving .'

**Reply: Done.**

p. 4 line 62/63: awkward wording: maybe rephrase it to 'Grounding has been suggested as a potential mechanism to affect the stability of MIT by delaying calving ( Massom…) .

**Reply: Done.**

p. 4 line 65-70: somewhere here a reference to Fig 3 would be useful.

**Reply: 'Fig 3' has been added after 'This accurate data set' in line ***.**

 p. 5 line 95: something wrong here in this sentence, 'fundamental' would be correct.

**Reply: Done.**

 p. 5 line 99: change to '…(DEM) for which the spatial coverage can be found in Figures 3b and 3c.'

**Reply: Done.**

p. 6 line 104: delete 'at least' as unclear how this is meant.

**Reply: Done.**

 p. 6 line 105: I do not understand this explanation, what do you mean by the bathymetric gap and how does identifying it help here? does the 2000 data extend furthest into the ice tongue? But this only really helps at the tip of the MIT (where it later grounds). Maybe delete this sentence.

**Reply: This sentence has been deleted.**

 p. 6 6 line 107. The data in west and east provides the outer boundaries (tie-points) for the interpolation but not really controls for seafloor depth underneath tongue!!!

**Reply: ', which provide control points for seafloor interpolation under the tongue' has been deleted.**

 p. 6. Line 109: awkward wording, maybe say: '…the MIT varies depending on distance to margin.'

**Reply: Done.**

p. 7 line 129 /130: awkward wording, maybe change to: ' The methods we designed for grounding detection of the MIT using ICESat/GLAS data are introduced here.'

**Reply: Done.**

p. 7 line 142: awkward wording, maybe change to: ' '…or high reflection from natural surfaces'.

**Reply: Done.**

P. 7 line 142: add a comma after 'Thus'

**Reply: Done.**

p. 7 line 147 '…for the instantaneous …' )not '…of the…'

**Reply: Done.**

p. 7 148/149: I do not understand this sentence: do you mean is 'available' for subsequent use?

**Reply: Yes.  'ready' has been replaced by 'available' in this sentence.**

 p. 8 line 167: should it not be 'from March 22' and 'from November 1'?

**Reply: Done.**

p. 9 line 179/180/181: awkward wording, I would simplify this to:

  '⋯where x and y are the horizontal positions directly from ICESat measurements, and X and Y the horizontal positions after relocation respectively. vx and vy are the horizontal components of the ice velocities.

**Reply: Done.**

 P. 9. Line 193: '…for November…' (not 'on November')

**Reply: Done.**

p. 10 line 197: I would introduce the 'lowest sea surface height symbol E_sea_level already here ((and not on line 219) and slightly rephrase and shorten this: '…and using the lowest sea surface height (E_sea_level) as reference for the sea surface elevation:' and then eqn (4)

**Reply: Done.**

 p. 10 line 200-202: there are a lot of 'the' missing here: '…where D is the ice draft…from the sea surface to the bottom of the ice; Hf is the freeboard, i.e. the vertical distance from the surface to the top of the snow; …are the densities…'

**Reply: Done.**

p. 10 line 203: '…FAC is the firn air content which corresponds to the decrease …'

**Reply: Done.**

p. 10 lines 206-209: should be clearer, rephrase to:

'The sea surface is taken as the lowest sea surface height (E_sealevel) and is derived from the minimum of all sea surface heights from the different ICESat/GLAS tracks between 2003 and 2009 and amounts in our case to -3.35m'.

**Reply: Done.**

p. 10 line 213: add a 'the' in front of 'ice bottom'.

**Reply: Done.**

p. 10 line 215/216: rephrase to: ' The elevation of the underside (bottom) of the tongue E_ice_bottom is calculated from:' then eqn (5)

**Reply: Done.**

p 11, lines 219 and 220 can then be removed (as repetition.

**Reply: Lines 219 and 220 have been removed.**

p. 11 line238: '…are available..' (not '…is available…') as it refers to measurements.

**Reply: Done.**

p. 246-249: this applies for any iceberg (not just for those calving from MIT) and awkward wording , I would change/simplify this to:

'…from surrounding icebergs that are slightly grounded under the assumption of hydrostatic equilibrium and known ice draft and freeboard. It is, however, critical to target and use icebergs that fulfil the condition of slight grounding.'

**Reply: Done.**

p. 12 line 258-260: maybe rephrase to: ' However, slowly drifting or nearly stationary icebergs in open water are good indicators for slight grounding and therefore be used to infer FAC.'

**Reply: Done.**

p. 13 line 266: I would rather use 'investigated' than 'identified' as you track/observe/study their position.

**Reply: Done.**

p. 13 line 269/270: awkward wording, maybe rephrase to:

'Fig. 4a shows that icebergs …..were almost stagnant and only slightly changed their positions and orientation over two months (…).'

**Reply: Done.**

p. 13 line 278: replace 'In this study,' by 'Therefore,' and 'employed' by 'used'

**Reply: Done.**

p. 13 line 282: 'where k refers to the icebergs 'A' or 'C',…'

**Reply: Done.**

p. 14 line 298: rephrase 'For Mertz we obtain a FAC of 4.87….. Other studies, using a time variable approach, modelled FAC values between 5 and 10 m (…) and in the absence of in-situ measurements our estimates seem consistent, but there are some shortcoming which should be .' Delete next two sentences (line 301-303.

**Reply: Done.**

p. 15 line 315: rephrase: '…may not refer to the same…'

**Reply: Done.**

p. 15 line 319: rephrase to ' …or observing a different portion of the iceberg…'

**Reply: Done.**

p. 15 line 322: '…a similar…' (not 'the similar')

**Reply: Done.**

p. 15 line 323: I would rather say '…for the inversion' (than 'to invert')

**Reply: Done.**

p. 16 line 337: I would leave away the 'Usually'

**Reply: Done.**

p. 16 line 338/339: delete the 'law' and say just '…by:' instead of ( 'by Eq. (9):'

**Reply: Done.**

p. 17 line 359: 'interpolated freeboard' (not 'freeboards')

**Reply: Done.**

p. 18 line 377: Rephrase to: 'Using Eq. (9) and kriging interpolation….'

**Reply: Done.**

p. 18 line 381: change to 'slight grounding' or 'slightly grounded'

**Reply: We have changed it to 'slight grounding'.**

p. p. 18: line 387/388: it is not clear to me what is really done with this buffer region and more important, how is E_diff calculated in this buffer region (where no surface data is available)? Clarify.

**Reply: More sentences have been added in Section 5. Now it reads 'Since the moving trajectory of the Mertz ice front changed by more than 40 degrees clockwise (Massom et al. 2015; Wang. 2014), a buffer region with radius of 2 km (region between black and grey lines in Fig. 6) is introduced to investigate grounding potential of the MIT. The freeboard in the buffer region is extrapolated using kriging interpolation method and the elevation difference is calculated.**

 p. 18 line 394: 'as illustrated in Table 2 and Fig 6…'

**Reply: Done.**

p. 18 line 395: 'was less than -23 m' (not 'were less…')

**Reply: Done.**

p. 18 line 396: replace 'From this point of view, we conclude that…' by 'This suggests that …'

**Reply: Done.**

p. 19 line 400: '…it would have been difficult…' (rather than 'it would be difficult'

**Reply: Done.**

p. 19 line 403: 'slight grounding' (or 'slightly grounded')

**Reply: We have changed it to 'slight grounding'.**

p. 19 line 405: again, 'strong grounding' (or 'strongly grounded')

**Reply: We change it to 'strong grounding'.**

p. 19 line 412: maybe 'tip' is better than 'flank'

**Reply: 'flank' is replaced by 'tip'.**

p. 19 line 415: 'For the grounded part (rather than 'grounding')

**Reply: Done.**

p. 19 line 418: '…lower right (northwest) section of the MIT…'

**Reply: Done.**

p. 20 line 430/431: I do not understand how a least square method is used to derive rate of area change, maybe the authors mean to derive the average trend of area change rate. Clarify.

**Reply: We have changed it to 'The average area-change trend of the MIT from 1989 to 2007 is also obtained using a least-squares method'.**

p. 20 line 434: 'surface behavior' is not the right term here, do you mean 'surface dynamics of the ice tongue'?.

**Reply: Yes. We have changed it to 'surface dynamics of the MIT.**

p. 21 line 445: '…would eventually have calved because of the effect of the shallow….'

**Reply: Done.**

p. 21 line 450: '…without considering an accidental such as the collision…'

**Reply: Done.**

p. 21 line 463: 'and the MIT calving cycle' and delete ', our explanation is'

**Reply: Done.**

p. 21 line 465-470: the wording of these new sentences is awkward, maybe change to:

'Variations in length of the MIT will prevent sea ice drifting from the east side to a variable degree. A long … because sea ice from the east side can not drift to the west side. The sea ice produced on the West side is blown seaward by the katabatic wind and thereby maintains a polynya and stable sea ice production. The sudden shortening of the MIT after a calving event therefore reduces ….'

**Reply: Done.**

p. 22 line 480/490: rephrase to: 'Additionally, the ice tongue continued to advance out into the ocean, where the bathymetry observation density is good.'

**Reply: Done.**

p. 22 line 488: rephrase to: '…since late 2002 is well supported by observations and which we take as evidence to infer the …'

**Reply: Done.**

p. 23 line 495: 'strong' (not 'strongly')

**Reply: Done.**

p. 23 line 498: '…as suggested by Massom…' (instead of 'pointed out')

**Reply: Done.**

p. 23 line 498/499: '…bathymetric data in the Mertz region…'

**Reply: Done.**

p. 23 line 503: '…around the Mertz…'

**Reply: Done.**

p. 23 line 504: '…understanding the MIT…'

**Reply: Done.**

p. 23 line 506: '… and is performing well.' (instead of 'is verified working well'.

**Reply: Done.**

p. 23 line 509: maybe 'dynamic behavior' is better than 'surface behavior'.

**Reply: 'dynamic behavior' has been used in this revision.**

p. 23 line 514: 'From these…'

**Reply: Done.**

p. 23 line 518: '… increasingly diverted by the obstructing seafloor shoal…'

**Reply: Done.**

p. 23 line 524: '… similar period for variations in sea surface conditions using…'seafloor sediment data. Thus, the shoaling on the seafloor combined with the rate of advance of the MIT determines the 70-year repeat cycle.'

**Reply: Done.**

Fig. 3 caption: line 685: shorten to: '… MIT from 2002 to 2008 marked with the colored polygons for different years.

**Reply: Done.**

It would be very useful to know from which years these bathymetric data in (b) and in (c) are.

**Reply: Unfortunately, we are not able to provide the detailed date for Figs 3b and 3c.**

Caption figure 6: please make a note here that no bathymetric data under most of ice tongue (for locations of bathymetric data see Fig 3b and c).

**Reply: 'Please note that no bathymetric data was available under most of the ice tongue and for locations of the bathymetric data, please refer to Figs 3b and 3c.' has been added.**

Table 1: two entries for C and second last column, make sure the minus sign is on same line as number. Similar, for Row B and last column.

**Reply: Done.**

Editor Andreas Vieli, 20 June 2016

[revised manuscript text omitted]

---

## Author Response (AR5)

**We are grateful to the Editor for his time and constructive comments to improve this manuscript. Here we address our reply point by point, in bold font.  All the Editor's comments are in regular font. All changes are marked with green and red in the revised manuscript.**

Editor Decision: Publish subject to technical corrections (26 Jul 2016) by Andreas Vieli

Comments to the Author:

Editors decision,

 Dear authors,

The 2nd re-revised version of the paper addresses carefully and well the editing-issues raised by the editor and clearly improved the quality and english language of the publication to an acceptable level. There are very few technical editing issues remaining which are listed below and when addressed the paper can be seen as accepted for publication in TC. I thank the authors for their latest effort in improving the paper.

Specific technical and very minor issues to be addressed:

p. 9, line 184: I think there should be a 'the' in front of 'freeboard map'.

**Reply: Done.**

p. 21 line 466: '…inside THE dash-dotted…'

**Reply: Done.**

p. 31 line 657: I think thie 'the ' in front of East Antarctica should be removed.

**Reply: Done.**

p. 36 line 689: awkward formulation '… can be clear…'. I would rather say '… can be read' or ' …over it which illustrates the density of the bathymetry measurements.'

**Reply: We have changed it into   ' …over it which illustrates the density of the bathymetry measurements.'**

p. 38 line 712/713: '…indicates the 7 km x 7km region used to investigate the accuracy of the kriging interpolation method.'

**Reply: Done.**

p. 41 line 743: remove 'that' after '…same legend as…'

**Reply: Done.**

p. 44 765/766: awkward wording, maybe change last subsentence to: '…and only includes those outside the 2000 Mertz boundary with an elevation difference less than 46 m.' (Is this what you mean here?).

**Reply: Yes. We have taken your advice and changed it into '···and only includes those outside 
[revised manuscript text omitted]